## Comment

quantum physics/applied mathematics

quantum correlations, local causality,
Bell's theorem, spinors, quaternions, octonions

**Author for correspondence:**
R. D. Gill
e-mail: gill@math.leidenuniv.nl

The accompanying can be viewed at http://doi.org/10.1098/rsos.180526.

# Comment on 'Quantum correlations are weaved by the spinors of the Euclidean primitives'

## R. D. Gill

Mathematical Institute, Leiden University, Leiden, The Netherlands

RDG, 0000-0001-5821-9986

I point out fundamental mathematical errors in the recent paper published in this journal 'Quantum correlations are weaved by the spinors of the Euclidean primitives' by Joy Christian.

## 1. Introduction

Christian [1], published in this journal *RSOS*, is perhaps the most ambitious of many papers by Joy Christian (in the sequel: 'the author') published between 2007 and 2021, all making the same assertion: the correlations predicted by quantum entanglement have a purely classical physical and 'locally realistic' explanation. In each paper, he proposes a *local hidden variables model* (but not always the same one) that according to him explains those correlations and reproduces them. According to the celebrated result known as Bell's theorem [2], this is impossible. Christian argues that Bell's proof of his theorem is mathematically wrong. In the paper discussed here, [1], he claims that the three-dimensional space around us globally has the geometry of the three-sphere $S^3$: the surface of the unit ball in $\mathbb{R}^4$. It is only locally flat. He connects this to special relativity, specifically to the solution of Einstein's field equations known as Friedmann–Robertson–Walker space–time with a constant spatial curvature. He furthermore connects it to the seven-sphere $S^7$, thought of as a quaternionic three-sphere rather than a real three-sphere.

A constant theme in the author's papers is to exploit a very well known and very established part of mathematics, *Geometric Algebra* (GA), based on the interplay between *Clifford Algebra* (CA), an important part of *abstract algebra*, and *Geometry*. It has been promoted as a universal language of physics. Important parts of quantum information theory have already been rewritten in the language of Geometric Algebra, though so far this did not have much impact on the field.

In Gill [3], I published a critique of his series of papers as it stood then. Actually, my paper was by no means the first paper refuting many of his claims, but the first to look at their development over the years, and almost the first one to appear in a peer-reviewed journal. The exception was Weatherall [4], published in *Foundations of Physics*. The author had not published any of his papers in peer reviewed journals, so many of the early critiques were not even submitted to regular journals, but remained, like his, as preprints on the preprint server arXiv.org. The author did respond vigorously to all critiques in yet more preprints and has always maintained that all criticism levelled at it had been mistaken and has been adequately refuted by himself.

In my paper, I argued that the author's work was built on a combination of ambiguous notation and elementary errors in logic, algebra and calculus. My paper included a section on an earlier version of the paper which is the topic of discussion here.

Time did not stand still, and after 2018, the author published a pair of companion papers, Christian (2019, 2020), in the journal *IEEE Access* [5,6]. At the invitation of the editors of that journal I published a 'Comment' Gill [7] on the second of the pair [6], and the author's (2021) 'Reply' [8] has also already appeared. Another invited 'Comment' by me [9] to the first of that pair has now also appeared in *IEEE Access*. I am grateful for the invitation by *RSOS* to react to [1] but in view of my earlier papers, I will keep this discussion very brief. I will focus on the mathematics, not on the physics, and refer to my earlier papers for the mathematical details. By the way, the algebraic core of the author's *RSOS* [1] paper was also analysed by Lasenby [10], who independently identified exactly the same problems which I had found.

If physicists want to 'rescue' the author's ideas because they see something in the underlying idea, it is up to them to do so. Physicists have again and again enriched mathematics by intuitively and with deep physical insight discovering patterns and abstract structures hitherto not known to mathematicians. This will surely happen again and again in the future. Debate on Bell's theorem will also continue for many years to come. In appendix A, I will discuss proposals by two referees of this paper to salvage the author's program. I argue that they cannot succeed. In fact, there are several other controversial and, in my opinion physically more interesting options for those who want to get around Bell's theorem. There are plenty of fora where these controversial alternative quantum foundations are vigorously discussed, and a huge (peer-reviewed) literature. I am afraid that the author's work has not added directly to the ongoing debate, though it did stimulate several original contributions which have made Bell's own case even stronger than it already was, and made the task of opponents thereby harder still.

In [8], the author stated

> *That is not to say that Bell's theorem does not have a sound mathematical core. When stated as a mathematical theorem in probability theory, there can be no doubt about its validity. My work on the subject does not challenge this mathematical core, if it is viewed as a piece of mathematics.*

This is however exactly what the author's works, and in particular his *RSOS* paper [1], do *not* do. He *does challenge the core mathematics* of Bell's work, and moreover, in [1], he builds this challenge on a new result of his own in pure mathematics (abstract algebra) which however is plain wrong. He went on in [8] to assert that what his work actually does is

> *challenge the metaphysical conclusions regarding locality and realism derived from that mathematical core. My work thus draws a sharp distinction between the mathematical core of Bell's theorem and the metaphysical conclusions derived from it.*

But drawing that 'essential distinction' is exactly what he does not do. He adopts Bell's own mathematical framing of the concepts of locality and realism and argues that one of the basic mathematical steps in Bell's proof is wrong. But it is his own argument which is evidently wrong. He does claim that our conception of space needs revision, but in view of its defects, his own work does not provide much support for that claim.

## 2. The Hurwitz theorem (algebra), Bell's theorem as probability theory and Bell's theorem as computer science

In this section, l discuss two established mathematical results which contradict *mathematical* claims in Christian [1]. These are: (a), the *Hurwitz theorem* (so called because it was conjectured by Hurwitz; it was only proved decades later) stating that $\mathbb{R}$, $\mathbb{C}$, $\mathbb{H}$ and $\mathbb{O}$ are the only four normed division algebras, see Baez [11]; and (b) the mathematical core of *Bell's theorem*, Bell [2] on the incompatibility of quantum mechanics (QM) with local realism (LR).

I also discuss two no-go theorems saying that certain quantum correlations cannot be simulated on a network of classical computers, where the allowed connections between computers mimic the spatial-temporal relations involved between physical subsystems of a typical Bell experiment. These can be thought of as theorems of computer science, but they are also merely repackagings of the mathematical core of Bell's theorem. I bring them up because the author illustrates his mathematical claims by Monte Carlo computer simulations, and I use them to provide more evidence for my own claims, that his theory is badly wrong.

These 'theorems from computer science' (specialism: distributed computing) are (c) *Gull's theorem*, Gull [12], Bell's core result proved in a beautiful and original way using Fourier theory, and (d) a theorem of my own published in Gill [13], which is actually Bell's core result *enhanced* using martingale theory (probability) and randomization (statistics).

Regarding Gull's theorem (c), astrophysicist Stephen Gull is another powerful promotor of geometric algebra. Readers familiar with Fourier series and time-series analysis might find Gull's proof of Bell's theorem much easier than Bell's. The fact that the result can be proven using a myriad different mathematical approaches is further evidence that it stands like a rock.

Regarding my own contribution (d), recent 'loophole-free experimental violations of Bell inequalities', [14] and others, have shown that it is possible to exhibit in the quantum optics laboratory phenomena that QM predicts but according to LR are impossible; phenomena which Einstein and others interpreted as 'spooky action at a distance'. This new generation of experiments is the most stringent ever. Alternative explanations of the observed correlations due to experimental shortcomings such as imperfect photo-detectors are ruled out. The problem is that experimental testing of Bell inequalities results in a finite amount of experimental data. Mathematical theorems about theoretical correlations in an ideal setting are not enough. My probabilistic result (d), in the form of an elegant refinement due to Hensen *et al.* [14], was used in the statistical analysis of the four celebrated loophole-free Bell experiments in 2015 and 2016 in Delft, Munich, Vienna and at NIST (Boulder, Colorado). Problems due to possible time variation, time trends, time dependence and finite sample size have been ruled out.

Computer simulations of Bell-type experiments have become very popular, and here too, the same statistical issues arise. They are popular both among opponents and among supporters of Bell. In the context of the author's work, some readers might imagine that thanks to mathematical subtleties which they cannot hope to appreciate, the author's work is still largely correct. An appeal to the Hurwitz theorem or to a mathematical version of Bell's theorem is an argument *ad verecundam*; by authority. Perhaps the author's work is merely blemished by details of notation or terminology. Fortunately for us, the author also presents computer simulations of his model. If that code faithfully represented his intended interpretation of his formulae, then by looking at the code one could deduce what he was trying to express in his published formulae. Moreover, one could check by inspection whether or not his code faithfully respects locality and realism. The simulations do seem to generate the results predicted by quantum mechanics. This means, in view of results (c) and/or (d), that it is not possible that the code satisfies the specifications agreed by Bell *and* the author.

It is indeed easy to check that the computer code effectively just draws the cosine curve built into the computer algebra package used by the author; it does not respect the constraints of local realism. It therefore weakens, rather than supports, his claims. His claims are unfounded, and the supporting evidence which he offers is actually further evidence that he has not succeeded in converting possible physical insight into hard science.

## 2.1. The Hurwitz theorem

The best reference for this theorem is Baez's award-winning paper [11] on the octonions, which starts with excellent expository material. According to Baez's theorem 1, the real numbers, the complex numbers, the quaternions and the octonions are the only normed division algebras (up to isomorphism, of course). Baez's definition of a normed division algebra is a real vector space endowed with a compatible (i.e. bilinear, i.e. satisfying distributivity axioms) multiplication operation which we call a product, and a norm making it a normed vector space, such that the norm is moreover multiplicative: the norm of a product is the product of the norms. The multiplication operation itself need not be commutative or even associative. (Commutativity is the requirement $ab = ba$, associativity is $a(bc) = (ab)c$). The useful and important thing to know here is that a division algebra has no zero divisors. There do not exist elements $A$ and $B$, neither equal to zero, such that $AB = 0$. However, the author's algebra has an element called the 'pseudo-scalar', I will denote it by $M$, such that $M^2 = 1$. It follows that $0 = M^2 - 1 = (M-1)(M+1)$. Taking norms, $0 = \|M-1\| \cdot \|M+1\|$. Hence

$\|M - 1\| = 0$ or $\|M + 1\| = 0$. Therefore $M - 1 = 0$ or $M + 1 = 0$, which implies that $M = 1$ or $M = -1$. That is a contradiction. The author's algebra is associative, the octonions are not. It is known as $Cl_{(0,3)}(\mathbb{R})$, the well studied even sub-algebra of $Cl_{(4,0)}(\mathbb{R})$.

A curious elementary mathematical error is that he defines two algebras, built from two eight-dimensional real vector spaces $\mathcal{K}^+$ and $\mathcal{K}^-$ by specifying a vector space basis for each algebra and multiplication tables for the eight basis elements of each algebra. But they are the *same* algebra. The linear spans of those two bases are trivially the same. The multiplication operation is the same. The author's claims regarding abstract algebra have naturally attracted the interest of algebraists. One of the founders of geometric algebra has published a paper, Lasenby [10], detailing the errors. The author, however, insists that his algebraic result is true and denies that it contradicts Hurwitz's theorem.

## 2.2. Bell's theorem: the mathematical core

Bell [2], as rapidly improved by Clauser, Horne, Shimony and Holt (1969) [15], essentially proves the following theorem. Suppose that $X_a$ and $Y_b$ are a family of random variables on a single probability space, taking values in the set $\{-1, +1\}$, and where $a$ and $b$ denote directions in ordinary three-dimensional Euclidean space, represented by unit vectors $a$, $b$. Then it is not possible that $\mathbb{E}(X_a Y_b) = -a \cdot b$ for all $a$ and $b$.

Bell's proof works by focussing on two choices for $a$ and two for $b$, delivering us four combinations of possible values of the pair $(a, b)$. Since four binary random variables are supported by a discrete probability space with just 16 elementary outcomes, a proof (using for instance the so-called CHSH inequality) can be framed in absolutely elementary terms [16]. No calculus is needed. No summation of infinite series. No knowledge of physics and in particular, no knowledge of quantum physics. (On a technical note: Bell's three correlation inequality is a corollary of the later four correlation inequality known as the CHSH-Bell inequality [15], which was later fully espoused by Bell himself.)

Bell used his mathematical result to argue that quantum mechanics violated the meta-physical principles of locality and realism. One can escape from this mathematical obstruction only by redefining the concepts of locality and realism. The author, however, does not take that well trodden route. He claims that the purely mathematical result is wrong. His alleged proof thereof is his explicit construction of a counterexample. I have elsewhere shown that his construction depends on simple errors in elementary algebra and calculus.

He also argues that Bell's proof contains a fundamental error in reasoning: the Bell-CHSH inequality involves correlations obtained from different sub-experiments involving measurements of non-commuting observables, and (he says) therefore cannot be combined. However, in quantum mechanics, even if two observables do not commute, a real linear combination of those observables is another observable. By the linearity encapsulated in the basic rules of quantum mechanics, expectation values of linear combinations of non-commuting observables are the same linear combination of the expectation values of each observable separately. If a local hidden variables model reproduces the statistical predictions of quantum mechanics, then it must reproduce this linearity.

## 2.3. Gull's theorem

Consider a computer simulation of a Bell-CHSH-type experiment. Initially, one could imagine three computers, a source computer sending information to two measurement locations, where two computers each simulate an apparatus which receives 'stuff' from a source, and a 'setting' (a measurement direction) supplied by an experimenter. The 'measurement station computers' each output an 'outcome' $\pm 1$. After a large number of trials, one collects the inputs (settings) and outputs (outcomes) together and computes the correlation (meaning in this context, just the mean value of the product) between the outcomes, for each possible pair of input settings.

Now, if that could be done, one could also create a copy of the source measurement station, including all the data which is stored on it at the beginning of the computer simulation, and then merge each of the copies of the source with the two measurement stations, giving us together two *completely separated* classical computers, which perform the following task.

The two computers are both loaded with data and a computer program, and then disconnected. After that, in $N$ rounds, each computer is supplied an input, and each computer supplies an output. After the complete run of $N$ 'trials' is completed, one collects all the data together, and correlates the outputs, for each possible pair of inputs. Gull [12] used to pose the question, as part of the 'Part II' (i.e. third year undergraduate) exam in Theoretical Physics at Cambridge University: is it possible to recover, in the

limit, the correlations $-a \cdot b$? He outlines a proof that this is impossible using an argument from Fourier analysis. Unfortunately he never published a formal proof, so we must make do with his overhead transparencies from a conference. Inspection of Gull's outline proof shows that he is making a particular 'no use of memory' assumption. Each of those computer's outputs, in trial $n$, depends only on the initial data stored in the computers and their programs, and on the new setting $a$ or $b$, and on the trial number $n$, but not on the previous $n-1$ inputs. Full mathematical details can be found in Gill & Karakozak [17].

The author's paper contains computer code: just one program which simulates many times two measurement settings and the value of some hidden variables which one can imagine created by nature in the source and transmitted to two measurement stations. Measurement outcomes are computed at each measurement station from the relevant setting and from the hidden variables. Next the program goes on to compute the correlation between measurement outcomes for any pair of measurement settings. This is where things go wrong: notice the code line `if (lambda==1) q=(NA NB) else q=(NB NA)`. One may check the algebra embodied in the code: the program simply computes $-a \cdot b$. It does not calculate the correlation between the actual ±1 valued measurement outcomes! The program does not implement the mathematical model given in the paper.

## 2.4. Probability bound on Bell-CHSH inequality

My paper, Gill [13], was my reaction to eminent scientists claiming that Bell's theorem was false, and some of them claiming to even be able to prove this by simulated computer models, running on a distributed network of computers. Some moreover claimed that Bell had not taken account of 'time' in his theory (not true, Bell discusses that explicitly in his famous 'Bertlmann's socks' paper, Bell [18]). I was concerned about inevitable statistical variation, and also by the possibility that the computer programs generating the $n$th pair of outcomes from the $n$th pair of settings might make use of information about the past $n-1$ settings and outcomes. Using martingale theory I proved a probability bound showing that deviations from the Bell-CHSH inequality of any size had exponentially small probability, provided only that the binary setting choices at each trial were performed, outside of each measurement station, and again and again, by two fair coin tosses. The measurement station computers were even allowed to communicate with one another and with the source *between* each trial. This result was later refined and improved and used by all the experimenters in the four famous 'loophole-free' Bell experiments of 2015, starting with the Delft experiment [14] and continuing with experiments in Vienna, at NIST in Boulder (Colorado), and in Munich. See [7] for further details.

## 3. Conclusion

The paper Christian [1] is irreparably flawed. The author wanted to provide theoretical underpinning to his physical intuition that quantum correlations are caused by the geometry of space, which, he suggests, is that of $S^3$, not of $\mathbb{R}^3$. He attempted to use geometric algebra for this task. I personally doubt that this is a fertile avenue for future research, but I am a mathematician, not a physicist. I would be delighted if anyone would prove me wrong. I do advise anyone interested in taking up the challenge to be well-informed as to the mathematical barriers embodied in the essential mathematical content at the core of Bell's theorem.

Data accessibility. This article has no additional data.
Authors' contributions. R.D.G.: conceptualization, formal analysis, investigation, writing—original draft, writing—review and editing.
Competing interests. I declare I have no competing interests.
Funding. R.D.G. received no funding for his research.
Acknowledgements. R.D.G. is grateful for the stimulating interactions with Dr Joy Christian over many years' lively debate.

## Appendix A

I here respond to comments of two referees.

A referee suggests that the violation of Bell's theorem in experiment shows that mathematics is inconsistent. According to him, Bell's theorem is both true and untrue. He points out problems with the ZFC axioms connected to infinite sets, and mentions that Einstein also intimated problems here

for physics. Of course it is in principle possible that the ZFC axioms are inconsistent. Gödel proved that consistency can never be proved. All this is however in my opinion irrelevant. My paper [16] presents a 'finitary' strengthening of the CHSH inequality—a discrete probability, finite sample size version. As I mentioned in the paper, Bell-CHSH uses two settings on each side of the experiment. The hidden variables can be reduced to the four binary counterfactual measurement outcomes. We need a discrete probability space with just $2^4 = 16$ outcomes. There is no way that deep logical issues concerning the existence of real numbers would spoil the *mathematical* theorems I have discussed in this paper.

There are, in my opinion, more interesting off-beat solutions, such as Tim Palmer's and Sabine Hossenfelder's and Gerard't Hooft's ideas based on superdeterminism. Palmer believes that fractal geometry and p-adic topology explain how nature can escape the constraints of Bell's theorem. Sabine Hossenfelder argues that there is nothing necessarily 'conspiratorial' in super-determinism; no need to delicately fine-tune parameters to make things come out exactly right. I think that she does not see the conspiratorial and unphysical ideas needed to imagine superdeterminism working its way all through a cascade of different kinds of random number generators used to generate measurement settings at two distant locations, in a subtle and perfect harmony with an underlying deterministic physics of lasers, photons and photo-dectors. I do not expect a satisfactory resolution of the problems facing those wanting to harmonize relativity and quantum theory and solve foundational issues in cosmology from this kind of approach, but it is good that these avenues are being explored.

Another referee suggests that the author's work is completely correct and that there are errors in my use of GA. This referee claims to have checked that the maths is correct using Matlab. Now it is true that many of the computations are locally correct. The referee says that the problem is that I do not acknowledge the author's new concepts of locality and realism and have not used his geometric setting. I disagree with this appraisal. The referee suggests that the only problem is that the author perhaps did not use entirely correct mathematical language. I suggest that the referee takes up the task of rewriting the author's model in unambiguous mathematical terms, and proceeds to implement the model faithfully in a Monte-Carlo computer simulation.

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
