## [Peer Review File · Royal Society Open Science]

Review History

RSOS-201909.R0 (Original submission)

Review form: Reviewer 1

Is the manuscript scientifically sound in its present form?

No

Are the interpretations and conclusions justified by the results?

Yes

Is the language acceptable?

No

Do you have any ethical concerns with this paper?

Yes

Have you any concerns about statistical analyses in this paper?

No

Recommendation?

Major revision is needed (please make suggestions in comments)

Comments to the Author(s)

Please read the attached report (see Appendix A).

Review form: Reviewer 2**Is the manuscript scientifically sound in its present form?**

Yes

Are the interpretations and conclusions justified by the results?

Yes

Is the language acceptable?

Yes

Do you have any ethical concerns with this paper?

No

Have you any concerns about statistical analyses in this paper?

No

Recommendation?

Accept with minor revision (please list in comments)

Comments to the Author(s)

State the mathematical facts keeping the discourse moderate and neutral.

Review form: Reviewer 3**Is the manuscript scientifically sound in its present form?**

Yes

Are the interpretations and conclusions justified by the results?

No

Is the language acceptable?

Yes

Do you have any ethical concerns with this paper?

No

Have you any concerns about statistical analyses in this paper?

No

Recommendation?

Major revision is needed (please make suggestions in comments)

Comments to the Author(s)

Bell's theorem has a sound mathematical core and when expressed as a theorem in probability theory, there can be no doubt about its truth.

Interestingly, when considered as a mathematical theorem about a probability distribution and its marginals, it can be derived without any reference to locality or realism.

More interestingly, (metaphysical) conclusions about the nature of locality and realism are often claimed to be the outcome of a mathematical theorem that does not even require either locality or realism for its derivation.

Also, there can several different ways in translating the concepts of locality and realism to mathematical conditions and Bell's definition is known to be not the unique and the only one.

The conclusions about the nature of locality and realism are then (forcefully) presented, even in some established literature, as exactly equivalent to the mathematical core of Bell's theorem.

In my understanding, Christian's work essentially shows that when equipped with an understanding of the geometry of space that is based on the mathematical formalism of geometric algebra, a local and realistic model can be developed for the correlations in a singlet state.

In other words, Bell's conclusions about the violation of his definition of local realism by the correlations in a singlet state is subject to the understanding of space itself.

In constructing his arguments, Christian does not challenge the mathematical core of Bell's theorem. However, it does challenge its version, its rhetoric, that puts the thrust of this theorem to be the metaphysical principles of locality and realism instead of its being a theorem from pure mathematics that even not requires either locality or realism for its mathematical derivation.

From this perspective, I see Christian's work drawing a clear line of separation between the mathematical core of Bell's theorem and (metaphysical) conclusions derived from it about the nature of locality and realism.

Christian's work requires us to place our focus strictly only on the mathematical core of Bell's theorem without invoking distractions and red herrings.

In my opinion, Gill's comment is misleading because it discredits Christian's work as well as his repute without making an effort on appreciating its real motivation.

Review form: Reviewer 4

Is the manuscript scientifically sound in its present form?

Yes

Are the interpretations and conclusions justified by the results?

Yes

Is the language acceptable?

Yes

Do you have any ethical concerns with this paper?

No

Have you any concerns about statistical analyses in this paper?

No

Recommendation?

Accept as is

Comments to the Author(s)

Dear Editor,

In this manuscript, R. Gill is concerned with refuting J. Christian's work against Bell's theorem published recently in RSOS. The core of debate is whether or not quantum mechanics is compatible with local realism. The debate is important for all working in this field. Although polemical, I suggest that the manuscript is published in its present form. Please find attached my review (see Appendix B).

Review form: Reviewer 5

Is the manuscript scientifically sound in its present form?

Yes

Are the interpretations and conclusions justified by the results?

Yes

Is the language acceptable?

Yes

Do you have any ethical concerns with this paper?

No

Have you any concerns about statistical analyses in this paper?

No

Recommendation?

Accept with minor revision (please list in comments)

Comments to the Author(s)

The article should absolutely be published. Errors in publications should be corrected in any reputable journal. The errors are clearly explained.

I would advise to maybe reduce the level of anger in the language.

Review form: Reviewer 6

Is the manuscript scientifically sound in its present form?

No

Are the interpretations and conclusions justified by the results?

No

Is the language acceptable?

Yes

Do you have any ethical concerns with this paper?

No

Have you any concerns about statistical analyses in this paper?

No

Recommendation?

Reject

Comments to the Author(s)

I have read this and related papers and I find that Gill repeats his criticisms against Christian's work. I won't go in a detailed analysis what Gill says and affirms, simply I will show his basic mistakes in GA computing, and why makes he this mistakes. Finally I understand the way as he thinks: he uses as way to model and compute vector calculus and matrix algebra and probabilistic theory to approach quantum mechanics. Christian made a significant step of analysing the problem of Bob and Aluce in an S^3 representation. Thanks to the topology of the quaternion (a subalgebra of 3D Euclidean geometric algebra) representation can explain the torsion of the positive oriented space (Pseudoscalar) for Bob (right hand rule) and the negative oriented space (Pseudoscalar) for Bob (right hand rule); that is why the hidden variable λ changes after a coin flip and since the orientation of the detectors are statistically independent, the well-known correlation $\cos(\theta_{A,B})^2$, is corroborated by the Christian's program. I wrote a Maple program and I get some results as Christian.

Even my undergraduate students taking a geometric algebra course will not make such a big mistake. Why? Because Gill is using Vector Calculus rules for multivector multiplication, which uses the geometric product:

Gills writes in this paper:

JC's algebra has an element called the "pseudo-scalar", I will denote it by M , such that $M^2 = 1$. It follows that $0 = M^2 - 1 = (M - 1)(M + 1)$. Taking norms, $0 = ||M - 1|| ||M + 1||$. Hence $||M + 1|| = 0$ or $||M - 1|| = 0$. Therefore $M - 1 = 0$ or $M + 1 = 0$, which implies that $M = 1$ or $M = -1$. That is a contradiction.

Now, Christian uses the Pseudo scalar I in the 3D Euclidean Geometric algebra, and I wrote a Maple program using G_3 and computed this and the results is -2 (negative due the signature of G_3 , i.e. $I^2 = -1$).

```
eCliffordversion();
```

```
dimension of the euclidean space
```

```
n := 3;
```

```
define bilinear form for  $R^{(n+1),2}$ 
```

```
p,q,r:=3,0,0;
```

$\text{dim_V}:=p+q+r;$

$\text{dim_Clpqr}:=2^{\text{dim_V}};$

$\text{Bsignature_pqr}:=\{p,q,r\};$

$\text{ecbas}:=\text{ecbasis}(\text{dim_V});$

Prepare the Pseudoscalar I_p , epsilon is a multivector basis for the scalar.

$I_p:=\text{ewedge}(e[1],e[2],e[3]); \text{coeff}(\text{ecmul}(I_p,I_p),\text{epsilon});$

Given the multivectors $U= (I_p-1)$ and $B=(I+1)$

The norm of U and V is $\sqrt{2}$.

As you see the pedantic comment of Gill's "That is a contradiction" is nonsense.

With respect to Bell's theorem, using otors (right hand) and the algebraic definitions of G_3 . I checked the equations of Christian and they seem correct. Here, Gill appears not really handle correctly basic definitions as geometric product, inner product, geometric product of bivectors and particularly the Pseudoscalar $= \lambda I$. And the role of the hidden variable λ (+-1), which changes the orientation of the Pseudoscalar in S^3 , so Bob events are pointing out to the north in S^3 and Alice to the south, this λ is computed with a uniform distribution random generator. I programmed G_3 in Maple, this is similar to the code of Christian and it works. What is really mistaken by Gill is to assert that Christian has taken the $1/2$ of inner product $a \cdot b = 1/2(ab+ba)$, as the probability. Christian did not say ever this. This is not true as independently of the inner product Christian throws the coin. In Gill's reformulation, he's tricking himself to get λ out of the a bivector product showing that λ is independent, $L(a; \lambda)L(b; \lambda) = \lambda^2 I^2 ab = -ab$. (3) So, in this way I think many equations of Christian are correct.

Decision letter (RSOS-201909.R0)

Dear Dr Gill

The Editors assigned to your paper RSOS-201909 "Comment on "Quantum correlations are weaved by the spinors of the Euclidean primitives"" have now received comments from reviewers and would like you to revise the paper in accordance with the reviewer comments and any comments from the Editors. Please note this decision does not guarantee eventual acceptance.

Please submit your revised manuscript and required files (see below) no later than 21 days from today's (ie 28-Apr-2021) date. Note: the ScholarOne system will 'lock' if submission of the revision is attempted 21 or more days after the deadline. If you do not think you will be able to meet this deadline please contact the editorial office immediately.

on behalf of Dr Derek Abbott (Associate Editor) and Miles Padgett (Subject Editor)
openscience@royalsociety.org

Associate Editor Comments to Author (Dr Derek Abbott):

Associate Editor: 1

Comments to the Author:

Editorial Comments

=====

- a) Please provide a detailed point by point response to all reviewer and editorial comments in your reply letter.
- b) The paper cannot be published in its present form. Please solely focus on the science and mathematics and remove all extraneous material.
- c) Remove all emotive language positive or negative, eg. "wonderful"
- d) Do not use titles such as "Dr" within the paper.
- e) Remove location information from the abstract and pointers about it. Only scientific pointers are permissible.
- f) Focus on dispassionate discourse.
- g) The ethics statement, disclaimer and competing interests statements are inappropriate in their

present form. Remove.

Reviewer comments to Author:

Reviewer: 1

Comments to the Author(s)

Please read the attached report.

Reviewer: 2

Comments to the Author(s)

State the mathematical facts keeping the discourse moderate and neutral.

Reviewer: 3

Comments to the Author(s)

Bell's theorem has a sound mathematical core and when expressed as a theorem in probability theory, there can be no doubt about its truth.

Interestingly, when considered as a mathematical theorem about a probability distribution and its marginals, it can be derived without any reference to locality or realism.

More interestingly, (metaphysical) conclusions about the nature of locality and realism are often claimed to be the outcome of a mathematical theorem that does not even require either locality or realism for its derivation.

Also, there can several different ways in translating the concepts of locality and realism to mathematical conditions and Bell's definition is known to be not the unique and the only one.

The conclusions about the nature of locality and realism are then (forcefully) presented, even in some established literature, as exactly equivalent to the mathematical core of Bell's theorem.

In my understanding, Christian's work essentially shows that when equipped with an understanding of the geometry of space that is based on the mathematical formalism of geometric algebra, a local and realistic model can be developed for the correlations in a singlet state.

In other words, Bell's conclusions about the violation of his definition of local realism by the correlations in a singlet state is subject to the understanding of space itself.

In constructing his arguments, Christian does not challenge the mathematical core of Bell's theorem. However, it does challenge its version, its rhetoric, that puts the thrust of this theorem to be the metaphysical principles of locality and realism instead of its being a theorem from pure mathematics that even not requires either locality or realism for its mathematical derivation.

From this perspective, I see Christian's work drawing a clear line of separation between the mathematical core of Bell's theorem and (metaphysical) conclusions derived from it about the nature of locality and realism.

Christian's work requires us to place our focus strictly only on the mathematical core of Bell's theorem without invoking distractions and red herrings.

In my opinion, Gill's comment is misleading because it discredits Christian's work as well as his repute without making an effort on appreciating its real motivation.

Reviewer: 4

Comments to the Author(s)

Dear Editor,

In this manuscript, R. Gill is concerned with refuting J. Christian's work against Bell's theorem published recently in RSOS. The core of debate is whether or not quantum mechanics is compatible with local realism. The debate is important for all working in this field. Although polemical, I suggest that the manuscript is published in its present form. Please find attached my review.

Reviewer: 5

Comments to the Author(s)

The article should absolutely be published. Errors in publications should be corrected in any reputable journal. The errors are clearly explained.

I would advise to maybe reduce the level of anger in the language.

Reviewer: 6

Comments to the Author(s)

I have read this and related papers and I find that Gill repeats his criticisms against Christian's work. I won't go in a detailed analysis what Gill says and affirms, simply I will show his basic mistakes in GA computing, and why makes he this mistakes. Finally I understand the way as he thinks: he uses as way to model and compute vector calculus and matrix algebra and probabilistic theory to approach quantum mechanics. Christian made a significant step of analysing the problem of Bob and Aluce in an S^3 representation. Thanks to the topology of the quaternion (a subalgebra of 3D Euclidean geometric algebra) representation can explain the torsion of the positive oriented space (Pseudoscalar) for Bob (right hand rule) and the negative oriented space (Pseudoscalar) for Bob (right hand rule); that is why the hidden variable λ changes after a coin flip and since the orientation of the detectors are statistically independent, the well-known correlation $\cos(\theta_{A,B})^2$, is corroborated by the Christian's program. I wrote a Maple program and I get some results as Christian.

Even my undergraduate students taking a geometric algebra course will not make such a big mistake. Why? Because Gill is using Vector Calculus rules for multivector multiplication, which uses the geometric product:

Gills writes in this paper:

JC's algebra has an element called the "pseudo-scalar", I will denote it by M , such that $M^2 = 1$. It follows that $0 = M^2 - 1 = (M - 1)(M + 1)$. Taking norms, $0 = ||M - 1|| ||M + 1||$. Hence $||M + 1|| = 0$ or $||M - 1|| = 0$. Therefore $M - 1 = 0$ or $M + 1 = 0$, which implies that $M = 1$ or $M = -1$. That is a contradiction.

Now, Christian uses the Pseudo scalar I in the 3D Euclidean Geometric algebra, and I wrote a Maple program using G_3 and computed this and the results is -2 (negative due the signature of G_3 , i.e. $I^2 = -1$).

```
eCliffordversion();
dimension of the euclidean space
n := 3;
define bilinear form for R^(n+1),2
p,q,r:=3,0,0;
dim_V:=p+q+r;
dim_Clqr:=2^dim_V;
Bsignature_pqr:=[p,q,r];
ecbas:=ebasis(dim_V);
```

Prepare the Pseudoscalar I_p , epsilon is a multivector basis for the scalar.

$I_p := \text{ewedge}(e[1], e[2], e[3]); \text{coeff}(\text{ecmul}(I_p, I_p), \text{epsilon});$

Given the multivectors $U = (I_p - 1)$ and $B = (I_p + 1)$

The norm of U and V is $\sqrt{2}$.

As you see the pedantic comment of Gill's "That is a contradiction" is nonsense.

With respect to Bell's theorem, using otors (right hand) and the algebraic definitions of G_3 . I checked the equations of Christian and they seem correct. Here, Gill appears not really handle correctly basic definitions as geometric product, inner product, geometric product of bivectors and particularly the Pseudoscalar $= \lambda I$. And the role of the hidden variable λ ($+1$), which changes the orientation of the Pseudoscalar in S^3 , so Bob events are pointing out to the north in S^3 and Alice to the south, this λ is computed with a uniform distribution random generator. I programmed G_3 in Maple, this is similar to the code of Christian and it works. What is really mistaken by Gill is to assert that Christian has taken the $1/2$ of inner product $a \cdot b = 1/2(ab + ba)$, as the probability. Christian did not say ever this. This is not true as independently of the inner product Christian throws the coin. In Gill's reformulation, he's tricking himself to get λ out of the a bivector product showing that λ is independent, $L(a; \lambda)L(b; \lambda) = \lambda^2 I^2 ab = -ab$. (3) So, in this way I think many equations of Christian are correct.

===PREPARING YOUR MANUSCRIPT===

If you have been asked to revise the written English in your submission as a condition of publication, you must do so, and you are expected to provide evidence that you have received language editing support. The journal would prefer that you use a professional language editing service and provide a certificate of editing, but a signed letter from a colleague who is a native speaker of English is acceptable. Note the journal has arranged a number of discounts for authors

using professional language editing services
(<https://royalsociety.org/journals/authors/benefits/language-editing/>).

===PREPARING YOUR REVISION IN SCHOLARONE===

<https://royalsociety.org/journals/authors/author-guidelines/#supplementary-material> to include a suitable title and informative caption. An example of appropriate titling and captioning may be found at https://figshare.com/articles/Table_S2_from_Is_there_a_trade-

off_between_peak_performance_and_performance_breadth_across_temperatures_for_aerobic_sc
ope_in_teleost_fishes_/3843624.

Author's Response to Decision Letter for (RSOS-201909.R0)

See Appendix C.

RSOS-201909.R1 (Revision)

Review form: Reviewer 1

Is the manuscript scientifically sound in its present form?

Yes

Are the interpretations and conclusions justified by the results?

Yes

Is the language acceptable?

No

Do you have any ethical concerns with this paper?

No

Have you any concerns about statistical analyses in this paper?

No

Recommendation?

Accept with minor revision (please list in comments)

Comments to the Author(s)

Please see attached PDF report (Appendix D).

Review form: Reviewer 2

Is the manuscript scientifically sound in its present form?

Yes

Are the interpretations and conclusions justified by the results?

Yes

Is the language acceptable?

Yes

Do you have any ethical concerns with this paper?

No

Have you any concerns about statistical analyses in this paper?

No

Recommendation?

Accept with minor revision (please list in comments)

Comments to the Author(s)

The present manuscript is a lucid and insightful piece of work. It clearly explains the mathematical core of Bell's inequality in its precise form. I concur with the author about Christian's errors in incorporating new definitions of locality and realism into a correct mathematical framework. The author has pointed out the algebra used in Christian's work is associative, but he should elucidate more on this point. Regarding the errors of the use of Geometric Algebra in the Christian's work, I concur with Lensby, and the author has provided sufficient literature for it. I agree with the author about the inappropriateness of Christian's claim.

Review form: Reviewer 3

Is the manuscript scientifically sound in its present form?

Yes

Are the interpretations and conclusions justified by the results?

Yes

Is the language acceptable?

Yes

Do you have any ethical concerns with this paper?

No

Have you any concerns about statistical analyses in this paper?

No

Recommendation?

Accept as is

Comments to the Author(s)

Thank you for reviewing your manuscript in view of my comments. As a valuable contribution to this subject, I recommend its publication.

Review form: Reviewer 4

Is the manuscript scientifically sound in its present form?

Yes

Are the interpretations and conclusions justified by the results?

No

Is the language acceptable?

Yes

Do you have any ethical concerns with this paper?

No

Have you any concerns about statistical analyses in this paper?

No

Recommendation?

Accept with minor revision (please list in comments)

Comments to the Author(s)

In the first round of review, I recommended the manuscript arguing that Bell and his followers as well as Bell-deniers are quite right in their beliefs as the so-called "Bell's theorem" in physics is, indeed, an undecidable hypothesis in mathematics. In my review I showed that the Copenhagen interpretation and its Born's rule leads to incompatibility and compatibility between quantum probability and classical probability at same time.

In this second round, the author included in his revision an Appendix with new arguments. I consider that the general aim of the debate about so-called "Bell's theorem" is a very valuable goal. I think, however, that the text needs to be improved before it is eventually published.

Firstly, I should point out that the author, in his revision, shuffled the reviews from the referees #1 and #4. This procedure is liable to mislead the readers since the reviews are a part of the debate. For a better understanding of the manuscript, I recommend that he corrects the mistakes in his revision.

That said, let me examine some of the arguments included in Appendix:

- The sentence "A referee [Referee 1] suggests that the violation of Bell's theorem ..." in Appendix [line 40] needs to be changed for "A referee [Referee 4] suggests that the violation of Bell's theorem ..."

- In Appendix, the author wrote "I am not an expert but I admit that it is possible that the ZFC axioms are inconsistent, though in my opinion not likely. All this is however in my opinion irrelevant."

The way he wrote suggests to us very strongly that his argument is passionate rather than scientific. These sentences are confusing, and it can harm the reading. How can the author say that "All this is however in my opinion irrelevant" if the author himself said that he is not an expert in the topic?

So, I suggest that the sentence "All this is however in my opinion irrelevant" must be suppressed to avoid cross purposes.

- The author also writes "I believe that this "foundations of mathematics" approaches is fruitless. It has been taken up a few times but gained no notable adherents."

These sentences must be rephrased because the so-called "Bell's theorem" has its substrate in Bell's inequalities, which are basic results from the Kolmogorov probability theory, which in turn is a result of set theory. As a consequence, Bell's theorem is directly linked to the foundations of

mathematics. And, it is also important to note that an approach for QM based on the foundations of mathematics was proposed by Albert Einstein in his last work, then, the author's belief that such an approach gained no notable adherents is quite controversial.

The author mentions that "There are much more interesting off-beat solutions, such as Tim Palmer's and Sabina Hossenfelder's and Gerard't Hooft's ideas based on superdeterminism."

It is worth highlighting here that the idea of superdeterminism is grounded on a possible loophole in Bell's theorem, i.e., the possibility of there being a local probabilistic model on the Boolean domain (CHSH at most 2, or not greater than 2) compatible with a nonlocal probabilistic model on the continuum (CHSH greater than 2). But it is interesting that whenever a local probabilistic model compatible with quantum probabilities is constructed, a probabilistic model incompatible with quantum probabilities is constructed at the same time. This is superdeterminism!

I recommend the manuscript for publication in RSOS, provided that the author considers the points listed above.

Review form: Reviewer 5

Is the manuscript scientifically sound in its present form?

Yes

Are the interpretations and conclusions justified by the results?

Yes

Is the language acceptable?

Yes

Do you have any ethical concerns with this paper?

No

Have you any concerns about statistical analyses in this paper?

No

Recommendation?

Accept with minor revision (please list in comments)

Comments to the Author(s)

It should be Sabine, not Sabina (a few times in the appendix).

Review form: Reviewer 6

Is the manuscript scientifically sound in its present form?

No

Are the interpretations and conclusions justified by the results?

No

Is the language acceptable?

No

Do you have any ethical concerns with this paper?

No

Have you any concerns about statistical analyses in this paper?

No

Recommendation?

Reject

Comments to the Author(s)

The paper doesn't contribute to knowledge. The author just criticizes and even in an offensive manner the Joi Christian work. It doesn't deserve to be published.

Decision letter (RSOS-201909.R1)

Dear Dr Gill

On behalf of the Editors, we are pleased to inform you that your Manuscript RSOS-201909.R1 "Comment on "Quantum correlations are weaved by the spinors of the Euclidean primitives"" has been accepted for publication in Royal Society Open Science subject to minor revision in accordance with the referees' reports. Please find the referees' comments along with any feedback from the Editors below my signature.

In addition to the comments provided by the reviewers, there are a number of additional modifications that the journal would like the author to make to their manuscript during revision, please, these are enumerated below:

1. Please can the usage of 'fatal' in the abstract be removed and instead replaced with 'fundamental' or 'serious' (or similar)? This will help make the Comment more dispassionate (see bullet 3 below).
2. In the second paragraph of the Comment's introduction, please could you include additional references to link to earlier works that have critiqued Joy Christian's work; at present the author says he is one of many with concerns, but it is not immediately which other works have provided this criticism, and it would be helpful to readers to include additional references to other works to provide additional context.
3. In correspondence with the editorial office, the Editors have asked us to encourage the author to moderate the tone of the manuscript, please. They expressed a concern that the manuscript as written remains more emotive than would ordinarily be expected from a Comment assessing the scientific and mathematical merits of the original paper. A suggestion made by the Editors that may help with this is reducing usage of the original author's name and instead referring to them as 'the author';
4. A number of references are made to 'George Lasenby' in the text, when it appears that the author intends to refer to 'Anthony Lasenby' instead - we encourage you to doublecheck this.

Please submit your revised manuscript and required files (see below) no later than 7 days from today's (ie 17-Sep-2021) date. Note: the ScholarOne system will 'lock' if submission of the revision is attempted 7 or more days after the deadline. If you do not think you will be able to meet this deadline please contact the editorial office immediately.

on behalf of Dr Derek Abbott (Associate Editor) and Miles Padgett (Subject Editor)
 openscience@royalsociety.org

Reviewer comments to Author:

Reviewer: 5

Comments to the Author(s)

It should be Sabine, not Sabina (a few times in the appendix).

Reviewer: 3

Comments to the Author(s)

Thank you for reviewing your manuscript in view of my comments. As a valuable contribution to this subject, I recommend its publication.

Reviewer: 2

Comments to the Author(s)

The present manuscript is a lucid and insightful piece of work. It clearly explains the mathematical core of Bell's inequality in its precise form. I concur with the author about Christian's errors in incorporating new definitions of locality and realism into a correct mathematical framework. The author has pointed out the algebra used in Christian's work is associative, but he should elucidate more on this point. Regarding the errors of the use of Geometric Algebra in the Christian's work, I concur with Lensby, and the author has provided sufficient literature for it. I agree with the author about the inappropriateness of Christian's claim.

Reviewer: 4

Comments to the Author(s)

In the first round of review, I recommended the manuscript arguing that Bell and his followers as well as Bell-deniers are quite right in their beliefs as the so-called "Bell's theorem" in physics is, indeed, an undecidable hypothesis in mathematics. In my review I showed that the Copenhagen interpretation and its Born's rule leads to incompatibility and compatibility between quantum probability and classical probability at same time.

In this second round, the author included in his revision an Appendix with new arguments. I consider that the general aim of the debate about so-called "Bell's theorem" is a very valuable goal. I think, however, that the text needs to be improved before it is eventually published.

Firstly, I should point out that the author, in his revision, shuffled the reviews from the referees #1 and #4. This procedure is liable to mislead the readers since the reviews are a part of the debate. For a better understanding of the manuscript, I recommend that he corrects the mistakes in his revision.

That said, let me examine some of the arguments included in Appendix:

- The sentence "A referee [Referee 1] suggests that the violation of Bell's theorem ..." in Appendix [line 40] needs to be changed for "A referee [Referee 4] suggests that the violation of Bell's theorem ..."

- In Appendix, the author wrote "I am not an expert but I admit that it is possible that the ZFC axioms are inconsistent, though in my opinion not likely. All this is however in my opinion irrelevant."

The way he wrote suggests to us very strongly that his argument is passionate rather than scientific. These sentences are confusing, and it can harm the reading. How can the author say that "All this is however in my opinion irrelevant" if the author himself said that he is not an expert in the topic?

So, I suggest that the sentence "All this is however in my opinion irrelevant" must be suppressed to avoid cross purposes.

- The author also writes "I believe that this "foundations of mathematics" approaches is fruitless. It has been taken up a few times but gained no notable adherents."

These sentences must be rephrased because the so-called "Bell's theorem" has its substrate in Bell's inequalities, which are basic results from the Kolmogorov probability theory, which in turn is a result of set theory. As a consequence, Bell's theorem is directly linked to the foundations of mathematics. And, it is also important to note that an approach for QM based on the foundations of mathematics was proposed by Albert Einstein in his last work, then, the author's belief that such an approach gained no notable adherents is quite controversial.

The author mentions that "There are much more interesting off-beat solutions, such as Tim Palmer's and Sabina Hossenfelder's and Gerard't Hooft's ideas based on superdeterminism."

It is worth highlighting here that the idea of superdeterminism is grounded on a possible loophole in Bell's theorem, i.e., the possibility of there being a local probabilistic model on the Boolean domain (CHSH at most 2, or not greater than 2) compatible with a nonlocal probabilistic model on the continuum (CHSH greater than 2). But it is interesting that whenever a local probabilistic model compatible with quantum probabilities is constructed, a probabilistic model incompatible with quantum probabilities is constructed at the same time. This is superdeterminism!

I recommend the manuscript for publication in RSOS, provided that the author considers the points listed above.

Reviewer: 6

Comments to the Author(s)

The paper doesn't contribute to knowledge. The author just criticizes and even in an offensive manner the Joi Christian work. It doesn't deserve to be published.

Reviewer: 1

Comments to the Author(s)
Please see attached PDF report.

===PREPARING YOUR MANUSCRIPT===

Your revised paper should include the changes requested by the referees and Editors of your manuscript. You should provide two versions of this manuscript and both versions must be provided in an editable format:
one version identifying all the changes that have been made (for instance, in coloured highlight, in bold text, or tracked changes);
a 'clean' version of the new manuscript that incorporates the changes made, but does not highlight them. This version will be used for typesetting.

===PREPARING YOUR REVISION IN SCHOLARONE===

At Step 3 'File upload' you should include the following files:
-- Your revised manuscript in editable file format (.doc, .docx, or .tex preferred). You should upload two versions:

- 1) One version identifying all the changes that have been made (for instance, in coloured highlight, in bold text, or tracked changes);
 - 2) A 'clean' version of the new manuscript that incorporates the changes made, but does not highlight them.
 - An individual file of each figure (EPS or print-quality PDF preferred [either format should be produced directly from original creation package], or original software format).
 - An editable file of each table (.doc, .docx, .xls, .xlsx, or .csv).
 - An editable file of all figure and table captions.
- Note: you may upload the figure, table, and caption files in a single Zip folder.
- Any electronic supplementary material (ESM).
 - If you are requesting a discretionary waiver for the article processing charge, the waiver form must be included at this step.
 - If you are providing image files for potential cover images, please upload these at this step, and inform the editorial office you have done so. You must hold the copyright to any image provided.
 - A copy of your point-by-point response to referees and Editors. This will expedite the preparation of your proof.

- Ensure that your data access statement meets the requirements at <https://royalsociety.org/journals/authors/author-guidelines/#data>. You should ensure that you cite the dataset in your reference list. If you have deposited data etc in the Dryad repository, please only include the 'For publication' link at this stage. You should remove the 'For review' link.
- If you are requesting an article processing charge waiver, you must select the relevant waiver option (if requesting a discretionary waiver, the form should have been uploaded at Step 3 'File upload' above).
- If you have uploaded ESM files, please ensure you follow the guidance at <https://royalsociety.org/journals/authors/author-guidelines/#supplementary-material> to include a suitable title and informative caption. An example of appropriate titling and captioning may be found at [https://figshare.com/articles/Table_S2_from_Is_there_a_trade-off_between_peak_performance_and_performance_breadth_across_temperatures_for_aerobic_sc ope_in_teleost_fishes_/3843624](https://figshare.com/articles/Table_S2_from_Is_there_a_trade-off_between_peak_performance_and_performance_breadth_across_temperatures_for_aerobic_scope_in_teleost_fishes_/3843624).

Author's Response to Decision Letter for (RSOS-201909.R1)

See Appendix E.

RSOS-201909.R2

Review form: Reviewer 1

Is the manuscript scientifically sound in its present form?

Yes

Are the interpretations and conclusions justified by the results?

Yes

Is the language acceptable?

Yes

Do you have any ethical concerns with this paper?

No

Have you any concerns about statistical analyses in this paper?

No

Recommendation?

Accept with minor revision (please list in comments)

Comments to the Author(s)

Please see one phrase highlighted and commented in the manuscript (Appendix F).

Review form: Reviewer 2

Is the manuscript scientifically sound in its present form?

Yes

Are the interpretations and conclusions justified by the results?

Yes

Is the language acceptable?

Yes

Do you have any ethical concerns with this paper?

No

Have you any concerns about statistical analyses in this paper?

No

Recommendation?

Accept as is

Comments to the Author(s)

The author has incorporated my suggestions as well as the concerns of other reviewers in the revised version. I recommend publication.

Review form: Reviewer 3

Is the manuscript scientifically sound in its present form?

Yes

Are the interpretations and conclusions justified by the results?

Yes

Is the language acceptable?

Yes

Do you have any ethical concerns with this paper?

No

Have you any concerns about statistical analyses in this paper?

No

Recommendation?

Accept as is

Comments to the Author(s)

N/A

Review form: Reviewer 4 (A de Castro)

Is the manuscript scientifically sound in its present form?

Yes

Are the interpretations and conclusions justified by the results?

Yes

Is the language acceptable?

Yes

Do you have any ethical concerns with this paper?

No

Have you any concerns about statistical analyses in this paper?

No

Recommendation?

Accept as is

Comments to the Author(s)

It is important to make clear for the broader public that so-called "Bell's theorem" states that a nonlocal model of probability - such as that given by square of sine function (Born rule) - has nothing to do with a local model of probability distribution. I showed this in detail in my report above for a case of three coins tossed simultaneously. So, the so-called "Bell's theorem" merely states that continuous sample spaces are incompatible with discrete (Boolean) sample spaces. [Just one remark about Born rule: considering the Copenhagen interpretation, the predictions of quantum mechanics can be given by the Born rule].

However, I also showed in my report that the Born rule can be directly obtained from a local model of probability distribution. Hence, the continuous sample space are compatible with discrete sample spaces.

That is a paradox, and means that proving or disproving anything about Bell's statement within ZFC is impossible. The theory has a loophole (superdeterminism), and as a result, the so-called "Bell's theorem" is, indeed, the very continuum hypothesis (an undecidable hypothesis), according to the references I mentioned in my report.

It's purely math!

As arguments and counterarguments regarding the so-called "Bell's theorem" are true, then I recommend the publication of the manuscript, once there cannot be an answer to the problem.

Review form: Reviewer 5

Is the manuscript scientifically sound in its present form?

Yes

Are the interpretations and conclusions justified by the results?

Yes

Is the language acceptable?

Yes

Do you have any ethical concerns with this paper?

No

Have you any concerns about statistical analyses in this paper?

No

Recommendation?

Accept with minor revision (please list in comments)

Comments to the Author(s)

The answers to the referees are lengthy and not so informative. I would suggest to shorten them or leave them away.

Decision letter (RSOS-201909.R2)

Dear Dr Gill

The Editors assigned to your paper RSOS-201909.R2 "Comment on "Quantum correlations are weaved by the spinors of the Euclidean primitives"" have now received comments from reviewers and would like you to revise the paper in accordance with the reviewer comments and any comments from the Editors. Please note this decision does not guarantee eventual acceptance.

Please submit your revisions by Friday 7 January 2022.

on behalf of Dr Derek Abbott (Associate Editor) and Miles Padgett (Subject Editor)
openscience@royalsociety.org

Associate Editor Comments to Author (Dr Derek Abbott):

Associate Editor: 1

Comments to the Author:

The paper contains a number of loose statements of the type that are verboten in a scholarly article. Please refer to the attachment called "RSOS-201909.R2_annotated.pdf". There you will find a pdf of your paper with detailed annotations for correction.

Associate Editor: 2

Comments to the Author:

(There are no comments.)

Reviewer comments to Author:

Reviewer: 1

Comments to the Author(s)

Please see one phrase highlighted and commented in the manuscript. (See attached "RSOS-201909.R2_with one comment on page 4.pdf")

Reviewer: 5

Comments to the Author(s)

The answers to the referees are lengthy and not so informative. I would suggest to shorten them or leave them away.

Reviewer: 2

Comments to the Author(s)

The author has incorporated my suggestions as well as the concerns of other reviewers in the revised version. I recommend publication.

Reviewer: 4

Comments to the Author(s)

It is important to make clear for the broader public that so-called "Bell's theorem" states that a nonlocal model of probability - such as that given by square of sine function (Born rule) - has nothing to do with a local model of probability distribution. I showed this in detail in my report above for a case of three coins tossed simultaneously. So, the so-called "Bell's theorem" merely states that continuous sample spaces are incompatible with discrete (Boolean) sample spaces. [Just one remark about Born rule: considering the Copenhagen interpretation, the predictions of quantum mechanics can be given by the Born rule].

However, I also showed in my report that the Born rule can be directly obtained from a local model of probability distribution. Hence, the continuous sample space are compatible with discrete sample spaces.

That is a paradox, and means that proving or disproving anything about Bell's statement within ZFC is impossible. The theory has a loophole (superdeterminism), and as a result, the so-called "Bell's theorem" is, indeed, the very continuum hypothesis (an undecidable hypothesis), according to the references I mentioned in my report.

It's purely math!

As arguments and counterarguments regarding the so-called "Bell's theorem" are true, then I recommend the publication of the manuscript, once there cannot be an answer to the problem.

Reviewer: 3

Comments to the Author(s)

N/A

===PREPARING YOUR MANUSCRIPT===

While not essential, it will speed up the preparation of your manuscript proof if accepted if you format your references/bibliography in Vancouver style (please see

<https://royalsociety.org/journals/authors/author-guidelines/#formatting>). You should include DOIs for as many of the references as possible.

If you have been asked to revise the written English in your submission as a condition of publication, you must do so, and you are expected to provide evidence that you have received language editing support. The journal would prefer that you use a professional language editing service and provide a certificate of editing, but a signed letter from a colleague who is a fluent speaker of English is acceptable. Note the journal has arranged a number of discounts for authors using professional language editing services (<https://royalsociety.org/journals/authors/benefits/language-editing/>).

===PREPARING YOUR REVISION IN SCHOLARONE===

Author's Response to Decision Letter for (RSOS-201909.R2)

See Appendix G.

RSOS-201909.R3

Review form: Reviewer 4 (A de Castro)

Is the manuscript scientifically sound in its present form?

Yes

Are the interpretations and conclusions justified by the results?

Yes

Is the language acceptable?

Yes

Do you have any ethical concerns with this paper?

Yes

Have you any concerns about statistical analyses in this paper?

No

Recommendation?

Accept as is

Comments to the Author(s)

Dear Editor,

In his response to decision letter, the author directly mentioned the Reviewer 4. Although the discussion has already gone on too long, I would like to make an additional comment, since I have been quoted. In case your decision is to publish the manuscript, I ask, please, that my comment is made available along with the review history.

Previously, I opted to accept the paper RSOS-201909, and I keep my recommendation to RSOS's editors, since it is important to consider arguments as well as counterarguments regarding the so-called Bell's theorem. It should be noted, however, that in his response to decision letter, the author states:

“Dear editors

Thank you very much for the positive review. I have carried out all the requests of the editors and the reviewers except for one referee, whose arguments I simply do not agree with. That is Reviewer 4, who has an utterly off-beat point of view, namely that Bell's theorem is both true and not true, and that mathematics as we know it is inconsistent [...].”

With respect to so-called Bell's theorem, it is worth pointing out that the author recently stated in another paper the what follows:

“It [Bell's theorem] cannot be explained by supposing the ZFC axioms (Zermelo-Fraenkel set theory with the axiom of choice) are inconsistent and have to be abandoned. The argument here is that the ZFC axioms lead to a contradiction - Bell's theorem.” [p. 9, Gill, R.D. Does Geometric Algebra Provide a Loophole to Bell's Theorem? Entropy 2020, 22, 61. DOI: 10.3390/e22010061]

If the author claimed, in a previous paper, that the so-called Bell's theorem is a contradiction, then the author should also agree, in his RSOS's manuscript, that the so-called Bell's theorem is both true and not true.

Another important point to be highlighted is that the author writes in his response to decision letter:

“I'm afraid I totally disagree with you [Reviewer 4]. I found your mathematical analysis unconvincing. I do not agree that there is some paradox here connected to inadequacy of ZFC [...].”

Well, in my mathematical analysis I showed that the ZFC axioms allow to prove, at same time, that the Born rule violates Bell's inequalities and that Born rule does not violate Bell's inequalities, which seems to be in line with the author in [p. 9, Gill, R.D. Does Geometric Algebra Provide a Loophole to Bell's Theorem? Entropy 2020, 22, 61. DOI: 10.3390/e22010061]:

“[...] The axioms therefore allow one both to prove a certain statement (the inequality) and its negation (violation of the inequality)”

As arithmetically shown in my report, the Born rule (nonlocal model of probability) is in a bijective relation with Bell's inequalities; however, there is enough experimental evidence that Bell's inequalities are incompatible with nonlocal models of probability. The existence of this true contradiction seems to produce a one-way bijection, which can be equivalent to the possibility of deterministic generation of “pseudo”-randomness (see in Levin, L. <https://www.cs.bu.edu/fac/lnd/expo/icm94.htm>). Considering the so-called Bell's theorem as an undecidable hypothesis is the basis for understanding the idea of superdeterminism.

That said, I think the author has lost himself in his arguments, but I keep my recommendation to RSOS's editors because believe that it is important debate.

Decision letter (RSOS-201909.R3)

Dear Dr Gill,

It is a pleasure to accept your manuscript entitled "Comment on "Quantum correlations are weaved by the spinors of the Euclidean primitives"" in its current form for publication in Royal Society Open Science. The comments of the reviewer(s) who reviewed your manuscript are included at the foot of this letter.

on behalf of Dr Derek Abbott (Associate Editor) and Miles Padgett (Subject Editor)
openscience@royalsociety.org

Associate Editor Comments to Author (Dr Derek Abbott):

Associate Editor: 1

Comments to the Author:

(There are no comments.)

Associate Editor: 2

Comments to the Author:

The paper is accepted contingent on three minor revisions that can be found in the attached annotated pdf of your paper.

Please check through your references and replace any arXiv references with the published versions, where they now exist.

Reviewer comments to Author:

Reviewer: 4

Comments to the Author(s)

Dear Editor,

In his response to decision letter, the author directly mentioned the Reviewer 4. Although the discussion has already gone on too long, I would like to make an additional comment, since I have been quoted. In case your decision is to publish the manuscript, I ask, please, that my comment is made available along with the review history.

Previously, I opted to accept the paper RSOS-201909, and I keep my recommendation to RSOS's editors, since it is important to consider arguments as well as counterarguments regarding the so-called Bell's theorem. It should be noted, however, that in his response to decision letter, the author states:

"Dear editors

Thank you very much for the positive review. I have carried out all the requests of the editors and the reviewers except for one referee, whose arguments I simply do not agree with. That is Reviewer 4, who has an utterly off-beat point of view, namely that Bell's theorem is both true and not true, and that mathematics as we know it is inconsistent [...]."

With respect to so-called Bell's theorem, it is worth pointing out that the author recently stated in another paper the what follows:

"It [Bell's theorem] cannot be explained by supposing the ZFC axioms (Zermelo-Fraenkel set theory with the axiom of choice) are inconsistent and have to be abandoned. The argument here is that the ZFC axioms lead to a contradiction - Bell's theorem." [p. 9, Gill, R.D. Does Geometric Algebra Provide a Loophole to Bell's Theorem? Entropy 2020, 22, 61. DOI: 10.3390/e22010061]

If the author claimed, in a previous paper, that the so-called Bell's theorem is a contradiction, then the author should also agree, in his RSOS's manuscript, that the so-called Bell's theorem is both true and not true.

Another important point to be highlighted is that the author writes in his response to decision letter:

"I'm afraid I totally disagree with you [Reviewer 4]. I found your mathematical analysis unconvincing. I do not agree that there is some paradox here connected to inadequacy of ZFC [...]"

Well, in my mathematical analysis I showed that the ZFC axioms allow to prove, at same time, that the Born rule violates Bell's inequalities and that Born rule does not violate Bell's inequalities, which seems to be in line with the author in [p. 9, Gill, R.D. Does Geometric Algebra Provide a Loophole to Bell's Theorem? Entropy 2020, 22, 61. DOI: 10.3390/e22010061]:

"[...] The axioms therefore allow one both to prove a certain statement (the inequality) and its negation (violation of the inequality)"

As arithmetically shown in my report, the Born rule (nonlocal model of probability) is in a bijective relation with Bell's inequalities; however, there is enough experimental evidence that Bell's inequalities are incompatible with nonlocal models of probability. The existence of this true contradiction seems to produce a one-way bijection, which can be equivalent to the possibility of deterministic generation of "pseudo"-randomness (see in Levin, L. <https://www.cs.bu.edu/fac/lnd/expo/icm94.htm>). Considering the so-called Bell's theorem as an undecidable hypothesis is the basis for understanding the idea of superdeterminism.

That said, I think the author has lost himself in his arguments, but I keep my recommendation to RSOS's editors because believe that it is important debate.

Appendix A

Report on Comment on “Quantum correlations are weaved by the spinors of the Euclidean primitives” by R.D. Gill

It is very laudable, that the author takes time to analyze errors in a published paper. And the arguments he brings forth appear valid. I therefore think, that in principle these comments also deserve to be published. That said, I still think that by making some adjustments his arguments can be even more objective and convincing. I therefore point out the following details:

1. Page 2, lines 17+18 Gill writes: “and cited in papers published in Nature and other prestigious journals.” This is an appeal to authority, which is fundamentally unscientific. I suggest to remove this.
2. Line 23: expression “a tangled web of nonsense”. This could be perceived as offensive, I would therefore also suggest to remove this expression. This would not diminish the scientific side of the argument.
3. Line 30: expression “No establishment conspiracy”. Unfortunately, the term “conspiracy” has during the time of the corona virus been used so often and it appears often just to abridge an otherwise interesting dialogue, that I would also prefer it to be dropped.
4. The next paragraph features expressions “ordinary crackpot”, “outrageous claims”, and “superfluous noise”. I suggest to remove these expressions, and replace them by clear objective non-polemic terms.
5. Lines 37+38 “respected pure mathematical journal Communications in Algebra”. By branding the journal as “respected” an implicit appeal to authority is made again, which I do not think is scientific. There is no journal that is beyond any form of human error.
6. Line 51: “almost none of his arXiv preprints got published.” That some publication channel does not publish the work of a certain author can have many reasons, including personal conflicts of interest between author and editors, scientifically correct but politically incorrect content, etc. I therefore cannot see the value of mentioning the judgement of an opaque review system of a preprint archive.
7. Page 3, lines 7+8: “are effectively absolving the scientific editors of scientific responsibility”. One may hold such an opinion, but I think to present the evidence upon which an editorial decision is based, is a worthy effort to bring transparency into a system which by its anonymity allows for unchecked unfairness.
8. Same paragraph: There is some truth to the criticism of the high price tags for open access publications. It actually means, who cannot afford it, or whose institution cannot afford it, will not get published, no matter how excellent the content of his article may be.
9. Gull and Gill’s theorem(s): I wonder who will really look up the references cited? Would it not be possible to fully state the theorem in technically clear language and then tell in which reference to find the full proof?
10. Paragraph (d): Expressions “eminent scientists” and “including one published in Nature”. The term “eminent” implies an unscientific value judgement, and “published in Nature” is at least an indirect appeal to authority. Just because something is published in this or that journal (be it Nature or RSOS or ...) does not make it more true or more objective. I therefore suggest to remove these terms, they do not contribute to the aim of the debate.
11. Paragraph (d): The style is highly inconsistent. Gill first sets out in third person, and in the last line suddenly switches to first person “my”. I advise to consistently write in third person only, or first person only.
12. Page 5, line 8, expression “disaster”. I do not share this outlook. No disaster has happened. A debate is unfolding, which in the end may help to understand the nature of Bell’s inequality better, similar to Einstein helping to clarify the meaning of uncertainty in quantum theory with his failed thought experiments. And there is the saying that science makes real progress by falsification. Successfully determining where J. Christian has erred, is progress in knowledge for him and for everybody who follows the debate. Claiming this to be a disaster discourages open debates in the future, which would not serve the progress of science in the long term.
13. I happily acknowledge the Acknowledgements. Otherwise by some of the terminology (like “ordinary crackpot”) I might have wrongly concluded that the author may have developed a personal resentment of J. Christian. I am glad to realize that this seems not to be the case.

Appendix B

In this manuscript, R. Gill is concerned with refuting J. Christian's work against Bell's theorem published recently in RSOS. The core of debate is whether or not quantum mechanics is compatible with local realism: J. Christian thinks so and R. Gill doesn't think so. Fundamentally, I must agree with R. Gill and J. Christian at the same time because the equipollent arguments of both authors are deducible from each other, and in this debate there is no winner. We are facing a dialetheism, where an affirmation and its negation are both true.

Bell's theorem states that the quantum probability and the classical probability are incompatible, which implies that continuous and discrete sample spaces have different (cardinalities) sizes. In accordance with Bell's theorem, nonlocal models of dependent events occurring on a continuous space conflict with the local models of independent events occurring on the boolean domain. Bell's intention was to reconcile quantum theory with the continuum hypothesis, as Einstein sought. In his last work, a note for the fifth edition (1955) of 'The Meaning of Relativity', Einstein wrote, "One can give good reasons why reality cannot at all be represented by a continuous field. From the quantum phenomena it appears to follow with certainty that a finite system of finite energy can be completely described by a finite set of numbers (quantum numbers). This does not seem to be in accordance with a continuum theory, and must lead to attempts to find a purely algebraic theory for the description of reality. However, nobody knows how to find the basis for such a description."

Nowadays, we can say that nobody knows how to find the basis for such a description because the problem is undecidable in ZFC. (K. Gödel and P. Cohen showed it in 1940 and 1963, respectively).

I will try, here, to contribute to the debate using Bell-Wigner's probabilistic approach ...

Consider the probability experiment of tossing three coins simultaneously, so that the throws are independent of each other. In such conditions, there are $2^3 = 8$ mutually exclusive events:

$$E_1 = \{HHH\},$$

$$E_2 = \{HHT\},$$

$$E_3 = \{HTH\},$$

$$E_4 = \{HTT\},$$

$$E_5 = \{THH\},$$

$$E_6 = \{THT\},$$

$$E_7 = \{TTH\},$$

$$E_8 = \{TTT\},$$

where H and T represent head and tail, respectively. Notably, the probabilities of occurring the disjoint events:

$$\{HT(H \text{ or } T)\},$$

$$\{H(H \text{ or } T)T\},$$

$$\{(H \text{ or } T)TH\},$$
 are as follows:

$$Pr[HT(H \text{ or } T)] = Pr[HTH] + Pr[HTT] - Pr[HTH \text{ and } HTT] = \frac{1}{8} + \frac{1}{8} = \frac{2}{8},$$

$$Pr[H(H \text{ or } T)T] = Pr[HHT] + Pr[HTT] - Pr[HHT \text{ and } HTT] = \frac{1}{8} + \frac{1}{8} = \frac{2}{8},$$

$$Pr[(H \text{ or } T)TH] = Pr[HTH] + Pr[TTH] - Pr[HTH \text{ and } TTH] = \frac{1}{8} + \frac{1}{8} = \frac{2}{8},$$

where, for mutually exclusive events:

$$Pr[HTH \text{ and } HTT] = Pr[HHT \text{ and } HTT] = Pr[HTH \text{ and } TTH] = 0$$

Given that the “golden” relation $\frac{2}{8} \leq \frac{2}{8} + \frac{2}{8}$ holds, then we can build the following Bell’s inequalities:

$$Pr[HT(H \text{ or } T)] \leq Pr[H(H \text{ or } T)T] + Pr[(H \text{ or } T)TH],$$

$$Pr[H(H \text{ or } T)T] \leq Pr[HT(H \text{ or } T)] + Pr[(H \text{ or } T)TH],$$

$$Pr[(H \text{ or } T)TH] \leq Pr[HT(H \text{ or } T)] + Pr[H(H \text{ or } T)T],$$

which are basic results of Kolmogorov's probability theory.

Now, take $H = (+)$ and $T = (-)$ in every event, so that (a.b.c) are the three independent throws:

Throws			Prob.
a	b	c	
H	H	H	1/8
H	H	T	1/8
H	T	H	1/8
H	T	T	1/8
T	H	H	1/8
T	H	T	1/8
T	T	H	1/8
T	T	T	1/8

 \Rightarrow

Throws		
a	b	c
+	+	+
+	+	-
+	-	+
+	-	-
-	+	+
-	+	-
-	-	+
-	-	-

As a consequence, we can write, for example, the Bell’s inequality

$$Pr[HT(H \text{ or } T)] \leq Pr[H(H \text{ or } T)T] + Pr[(H \text{ or } T)TH]$$

in this form:

$$\frac{\mathcal{P}r \left| \begin{array}{ccc} a & b & c \\ H & T & (H \text{ or } T) \end{array} \right.}{\mathcal{P}r[a+, b-]} \leq \frac{\mathcal{P}r \left| \begin{array}{ccc} a & b & c \\ H & (H \text{ or } T) & T \end{array} \right.}{\mathcal{P}r[a+, c-]} + \frac{\mathcal{P}r \left| \begin{array}{ccc} a & b & c \\ (H \text{ or } T) & T & H \end{array} \right.}{\mathcal{P}r[b-, c+]}$$

where, the logical disjunction ($H \text{ or } T$) doesn't matter.

Notice that a_{\pm} , b_{\pm} , c_{\pm} are arrays storing data about throws and measurements. These data structures are, therefore, vectors identified by one array index \pm on a 2-d space.

Let the vectors be orthonormal in an inner product space:

So, we can write the following trigonometric identities:

$$\langle b-|a+ \rangle = \cos(45^\circ + 90^\circ) = \sin 45^\circ$$

$$\langle c-|a+ \rangle = \cos(22.5^\circ + 90^\circ) = \sin 22.5^\circ$$

$$\langle c+|b- \rangle = \cos(22.5^\circ + 90^\circ) = \sin 22.5^\circ$$

As one can define a probability amplitude as a sinusoidal function, then sine-squared function is a probability measure.

Thus, we have the quantum probabilities:

$$Pr[a+, b-] = \langle b-|a+ \rangle^2 = \sin^2 45^\circ$$

$$Pr[a+, c-] = \langle c-|a+ \rangle^2 = \sin^2 22.5^\circ$$

$$Pr[b-, c+] = \langle c+|b- \rangle^2 = \sin^2 22.5^\circ$$

As a result, the inequality:

$$Pr[a+, b-] \leq Pr[a+, c-] + Pr[b-, c+]$$

Stays:

$$\sin^2 45^\circ \leq \sin^2 22.5^\circ + \sin^2 22.5^\circ,$$

or equivalently,

$$\sin^2 45^\circ \leq 2\sin^2 22.5^\circ,$$

or yet as the "red" relation:

$$\frac{1}{2}\sin^2 45^\circ \leq \frac{1}{2}\sin^2 22.5^\circ + \frac{1}{2}\sin^2 22.5^\circ,$$

which is a violation¹, because $0.2500 \leq 0.1464$. This implies that quantum probability, therefore, conflicts with classical probability!!!

¹ J.S.Bell used this violation to prove his theorem in "Bertlmann's socks and the nature of reality", *Journal de Physique*, Colloque C2, suppl. au numero 3, Tome 42 (1981) pp C2 41-61, reproduced as Ch. 16 of *Speakable and Unsayable in Quantum Mechanics*, Cambridge University Press 1987.

But on the other hand, we must remember that the aforementioned “golden” relation:

$$\frac{2}{8} \leq \frac{2}{8} + \frac{2}{8}$$

is equivalent to inequality $1 \leq 1 + 1$, and, consequently, it can be also written as $1 - 1 \leq 1$.

Or yet as:

$$1 - (1 + 0) \leq 1 + 0$$

Taking the absolute value of both sides, we have $|1 - (1 + 0)| \leq |1 + 0|$, which can be also rewritten as:

$$1 - |1 + 0| \leq |1 + 0|$$

Or as:

$$1 - \frac{1}{|1+0|} \leq \frac{1}{|1+0|},$$

because $|1 + 0| = \frac{1}{|1+0|}$.

Notice also that $\frac{1}{|1+0|} = \left[\frac{1}{|1+0|} \right]^2$. So, $1 - \frac{1}{|1+0|} \leq \left[\frac{1}{|1+0|} \right]^2$, and writing the integer 1_{10} in decimal system as $(0,1)_2$ in the binary system, we have:

$$1 - \frac{1}{|(0,1)+(1,0)|} \leq \left[\frac{1}{|(0,1)+(1,0)|} \right]^2,$$

since integer $0_{10} = NOT(1_{10}) = (1,0)_2$ is congruent to $2 \pmod 2$. The Boolean values 0 and 1 correspond to mutually orthogonal arrays typically written as two-component complex-valued spinors: the computational basis for a single quantum bit.

As a result,

$$1 - \frac{1}{|(1,1)|} \leq \left[\frac{1}{|(1,1)|} \right]^2$$

Remembering that the absolute value of a vector is equal to the square root of the sum of the squares of its components, we have that:

$$1 - \frac{1}{\sqrt{1^2+1^2}} \leq \left[\frac{1}{\sqrt{1^2+1^2}} \right]^2$$

Or,

$$1 - \frac{1}{\sqrt{2}} \leq \left[\frac{1}{\sqrt{2}} \right]^2$$

This inequality can be also written in the trigonometric form: $1 - \sin 45^0 \leq \sin^2 45^0$, because $\sin 45^0 = \frac{1}{\sqrt{2}}$.

And, considering the trigonometric identity $1 - \sin 45^0 = 2\sin^2 22.5^0$, we then can reformulate the inequality as follows:

$$2\sin^2 22.5^0 \leq \sin^2 45^0$$

Or yet as the "green" relation:

$$\frac{1}{2}\sin^2 22.5^\circ + \frac{1}{2}\sin^2 22.5^\circ \leq \frac{1}{2}\sin^2 45^\circ,$$

which is not a violation, because $0.1464 \leq 0.2500$. This implies that quantum probability, therefore, does not conflict with classical probability!!!

The relations "red" and "green", showed, here, that Bell's theorem in physics is an undecidable problem in mathematics. Bell's idea was to show that the continuous space and the discrete space have different sizes. Nowadays, however, we know that the continuum hypothesis cannot be proved nor disproved in Zermelo–Fraenkel set theory with the axiom of choice included (ZFC). This means that there is an impassable barrier for us to show whether quantum physics is incompatible with local realism or not.

Appendix C

Associate Editor

a) Please provide a detailed point by point response to all reviewer and editorial comments in your reply letter.

RDG: Done

b) The paper cannot be published in its present form. Please solely focus on the science and mathematics and remove all extraneous material.

RDG: Done.

c) Remove all emotive language positive or negative, eg. "wonderful"

RDG: Done

d) Do not use titles such as "Dr" within the paper.

RDG: Done

e) Remove location information from the abstract and pointers about it. Only scientific pointers are permissible.

RDG: Done

f) Focus on dispassionate discourse.

RDG: Done

g) The ethics statement, disclaimer and competing interests statements are inappropriate in their present form. Remove.

RDG: Done

Reviewer 1

Reviewer 1's report consists mainly of his own proof of Bell's 1964 inequality, leading up to the assertion: "Bell's theorem in physics is an undecidable problem in mathematics. Bell's idea was to show that the continuous space and the discrete space have different sizes. Nowadays, however, we know that the continuum hypothesis cannot be proved nor disproved in Zermelo–Fraenkel set theory with the axiom of choice included (ZFC). This means that there is an impassable barrier for us to show whether quantum physics is incompatible with local realism or not.

RDG: I totally disagree with your conclusion. Nowadays, we have proofs of Bell's theorem which rely on simple finite counting methods. This was already argued in the paper and now I have added an explicit discussion of your point of view in a new appendix. I also point out that there are a number of other definitely controversial ways to escape from Bell's conclusions (super determinism; new definitions of locality and/or of realism) which in my opinion are more worthy of consideration. I am prepared to add a reference to a paper which works out your point of view if that is allowed by the editors and if you are able to suggest a suitable reference. I did not go looking for one myself because I want to keep the paper short and on target.

Reviewer 2

State the mathematical facts keeping the discourse moderate and neutral.

RDG: Done

Reviewer 3

Bell's theorem has a sound mathematical core and when expressed as a theorem in probability theory, there can be no doubt about its truth.

Interestingly, when considered as a mathematical theorem about a probability distribution and its marginals, it can be derived without any reference to locality or realism.

More interestingly, (metaphysical) conclusions about the nature of locality and realism are often claimed to be the outcome of a mathematical theorem that does not even require either locality or realism for its derivation.

Also, there can several different ways in translating the concepts of locality and realism to mathematical conditions and Bell's definition is known to be not the unique and the only one.

The conclusions about the nature of locality and realism are then (forcefully) presented, even in some established literature, as exactly equivalent to the mathematical core of Bell's theorem.

In my understanding, Christian's work essentially shows that when equipped with an understanding of the geometry of space that is based on the mathematical formalism of geometric algebra, a local and realistic model can be developed for the correlations in a singlet state.

In other words, Bell's conclusions about the violation of his definition of local realism by the correlations in a singlet state is subject to the understanding of space itself.

In constructing his arguments, Christian does not challenge the mathematical core of Bell's theorem.

However, it does challenge its version, its rhetoric, that puts the thrust of this theorem to be the metaphysical principles of locality and realism instead of its being a theorem from pure mathematics that even not requires either locality or realism for its mathematical derivation.

From this perspective, I see Christian's work drawing a clear line of separation between the mathematical core of Bell's theorem and (metaphysical) conclusions derived from it about the nature of locality and realism.

Christian's work requires us to place our focus strictly only on the mathematical core of Bell's theorem without invoking distractions and red herrings.

In my opinion, Gill's comment is misleading because it discredits Christian's work as well as his repute without making an effort on appreciating its real motivation.

RDG: Of course the sound mathematical core of Bell's theorem can be derived without reference to any concepts from physics. Mathematics is mathematics.

Metaphysical statements about the nature of locality and realism need to adhere to normal standards of logic and to respect mathematical truths.

Yes, there are many ways to translate the concepts of locality and realism into mathematical physics, and it is very well known that Bell's is not the only one. Bell did attempt to formulate those concepts in a way which Einstein would have appreciated. Note that in the famous EPR paper (E = Einstein) a conception of realism is actually deduced by supposing that certain QM predictions are correct and by assuming a modest notion of locality. Now the kernel of your criticism is the following claim: *In my understanding, Christian's work essentially shows that when equipped with an understanding of the geometry of space that is based on the mathematical formalism of geometric algebra, a local and realistic model can be developed for the correlations in a singlet state.* Certainly, this was Christian's aim. My work shows that he fails in this endeavour, because of mathematical mistakes. He repeatedly claims to have disproved mathematical theorems by providing *mathematical counter-examples*, and he also claims to show where Bell's *mathematical reasoning* goes mathematically astray. The reviewer goes on to say: *Bell's conclusions about the violation of his definition of local realism by the correlations in a singlet state is subject to the understanding of space itself.* I disagree, and anyway, Christian's work is so mathematically flawed that it hardly provides any support for claims such as that of the reviewer. Christian's work utterly fails in drawing a clear line of separation between mathematical theory and metaphysical conclusions drawn from it! Christian's work is a distraction and red herring from the fascinating and important issues which Bell brought to the forefront. The reviewer concludes *Gill's comment is misleading because it discredits Christian's work as well as his repute without making an effort on appreciating its real motivation.* This is not true. I do appreciate Christian's real motivation. Christian's repute has unfortunately been severely damaged by his own work. Many persons have repeatedly pointed out mathematical errors. Christian never acknowledges them, but he does write new papers in which the same errors are still made but are less visible.

He repeatedly changes his mathematical claims while always claiming they never changed. This has caused damage to his repute.

Reviewer 4

In this manuscript, R. Gill is concerned with refuting J. Christian's work against Bell's theorem published recently in RSOS. The core of debate is whether or not quantum mechanics is compatible with local realism. The debate is important for all working in this field. Although polemical, I suggest that the manuscript is published in its present form. It is very laudable, that the author takes time to analyse errors in a published paper. And the arguments he brings forth appear valid. I therefore think, that in principle these comments also deserve to be published. That said, I still think that by making some adjustments his arguments can be even more objective and convincing. I therefore point out the following details.

1. Page 2, lines 17+18 Gill writes: "and cited in papers published in Nature and other prestigious journals." This is an appeal to authority, which is fundamentally unscientific. I suggest to remove this.

RDG: Done

2. Line 23: expression "a tangled web of nonsense". This could be perceived as offensive, I would therefore also suggest to remove this expression. This would not diminish the scientific side of the argument.

RDG: Done

3. Line 30: expression "No establishment conspiracy". Unfortunately, the term "conspiracy" has during the time of the corona virus been used so often and it appears often just to abridge an otherwise interesting dialogue, that I would also prefer it to be dropped.

RDG: Many of those who believe Bell was wrong argue that "the establishment" is suppressing their work. Christian is one such person. This is why the idea of the quantum Randi challenge was invented and published by Serge Vongehr some years ago. In a debate with such persons urge them to self-publish a computer simulation model on internet, and promote it through social media, so that thanks to internet everyone is in principle able to verify whether or not they have succeeded in disproving Bell's theorem. This is why Christian's recent papers do include simulation examples. That is nice because they provide an alternative route to seeing that his work is just nonsense. Hence I do still use the word, just once, in the revised manuscript, where I explain the role of simulation models in this field.

4. The next paragraph features expressions "ordinary crackpot", "outrageous claims", and "superfluous noise". I suggest to remove these expressions, and replace them by clear objective non-polemical terms.

RDG: Done

5. Lines 37+38 "respected pure mathematical journal Communications in Algebra". By branding the journal as "respected" an implicit appeal to authority is made again, which I do not think is scientific. There is no journal that is beyond any form of human error.

RDG: Done

6. Line 51: "almost none of his arXiv preprints got published." That some publication channel does not publish the work of a certain author can have many reasons, including personal conflicts of interest between author and editors, scientifically correct but politically incorrect content, etc. I therefore cannot see the value of mentioning the judgement of an opaque review system of a preprint archive.

RDG: The referee misunderstands me. Christian has successfully "published" a very long list of these papers on arXiv. Till recently however, none of them got published in peer-reviewed journals. Only recently has he at last found a way to get his ideas published in what should be high profile peer-reviewed journals (IEEE Access, and Royal Society Open Science).

7. Page 3, lines 7+8: “are effectively absolving the scientific editors of scientific responsibility”. One may hold such an opinion, but I think to present the evidence upon which an editorial decision is based, is a worthy effort to bring transparency into a system which by its anonymity allows for unchecked unfairness.

RDG: I have removed all discussion of currently fashionable editorial practices

8. Same paragraph: There is some truth to the criticism of the high price tags for open access publications. It actually means, who cannot afford it, or whose institution cannot afford it, will not get published, no matter how excellent the content of his article may be.

RDG: Yes I agree, but anyway, I have removed all discussion of currently fashionable science publishing practices

9. Gull and Gill’s theorem(s): I wonder who will really look up the references cited? Would it not be possible to fully state the theorem in technically clear language and then tell in which reference to find the full proof?

RDG: It certainly would be possible to state some theorems explicitly (which requires definitions, notation, references) but I have recently done exactly that in other published papers, where I also give what I consider the necessary elucidation of those theorems. I wanted to keep this paper as brief as possible. But if the editors will allow/invite me to make my paper longer still, I can do this.

10. Paragraph (d): Expressions “eminent scientists” and “including one published in Nature”. The term “eminent” implies an unscientific value judgement, and “published in Nature” is at least an indirect appeal to authority. Just because something is published in this or that journal (be it Nature or RSOS or ...) does not make it more true or more objective. I therefore suggest to remove these terms, they do not contribute to the aim of the debate.

RDG: Done

11. Paragraph (d): The style is highly inconsistent. Gill first sets out in third person, and in the last line suddenly switches to first person “my”. I advise to consistently write in third person only, or first person only.

RDG. Thanks, I hope I have fixed this now.

12. Page 5, line 8, expression “disaster”. I do not share this outlook. No disaster has happened. A debate is unfolding, which in the end may help to understand the nature of Bell’s inequality better, similar to Einstein helping to clarify the meaning of uncertainty in quantum theory with his failed thought experiments. And there is the saying that science makes real progress by falsification. Successfully determining where J. Christian has erred, is progress in knowledge for him and for everybody who follows the debate. Claiming this to be a disaster discourages open debates in the future, which would not serve the progress of science in the long term.

RDG: Well, I think there are some real debates going on concerning the foundations of quantum mechanics, and the work of Christian published in high profile journals impedes them by adding a lot of *noise*, and it distracts from them. But anyway: I have removed such emotive language.

13. I happily acknowledge the Acknowledgements. Otherwise by some of the terminology (like “ordinary crackpot”) I might have wrongly concluded that the author may have developed a personal resentment of J. Christian. I am glad to realize that this seems not to be the case.

RDG: I both like J. Christian quite a lot, and admire his obvious talents, *and* consider him an ordinary crackpot. I have been at a workshop with him, and I have met him in Oxford and discussed his work with him. I have no personal resentment of him at this time, though in the past I found many of his public comments about me very offensive (they can be easily found on internet! How do you like: “not a physicist,

not even a mathematician: merely a third rate statistician". He even credits some famous scientists for this characterisation). I was also very annoyed that he stole some mathematics from me and then published it as his own while at the same time misrepresenting the content. Human beings are very complex . They moreover change over the years. I am interested in psychology and sociology, so to sum up: I find his work fascinating, at a meta-level: how can an intelligent and educated person become so earnestly devoted to such crazy ideas, and perhaps more importantly, how on earth can it be, that other scientists do not see through them? I think there is much to learn from studying the publication of these papers. But anyway: I leave the psychology and sociology of this extraordinary case out of this paper.

Reviewer 5

The article should absolutely be published. Errors in publications should be corrected in any reputable journal. The errors are clearly explained. I would advise to maybe reduce the level of anger in the language.

RDG: Thank you! I have followed your advice.

Reviewer 6

I have read this and related papers and I find that Gill repeats his criticisms against Christian's work. I won't go into a detailed analysis of what Gill says and affirms, I will simply show his basic mistakes in GA computing, and why he makes these mistakes. Finally I understand the way he thinks: he uses as way to model and compute [classical] vector calculus and matrix algebra and probabilistic theory to approach quantum mechanics. ... My undergraduate students taking a geometric algebra course will not make such a big mistake. Why? Because Gill is using Vector Calculus rules for multivector multiplication, which uses the geometric product.

RDG: I do not use classical vector calculus and matrix algebra and probability theory to approach quantum mechanics. My paper is not about quantum mechanics at all. It is about mathematical mistakes in Christian's paper. The reviewer writes about his own personal interpretation of a small part of what Christian has written.

I wrote a Maple program using G_3 and computed this and the result is -2 (negative due the signature of G_3 , i.e. $I^2=-1$) ... As you see the pedantic comment of Gill's "That is a contradiction" is nonsense.

RDG: the reviewer has rewritten a fragment of Christian's GA code and gets, of course, the same answer. Christian's program computes the cosine function in a somewhat roundabout way. It does not faithfully implement his mathematical model at all.

What is really mistaken by Gill is to assert that Christian has taken the $1/2$ of inner product $a \cdot b = 1/2(ab+ba)$, as the probability. Christian did not say ever this.

RDG: I did not say that Christian says this. I say that that is effectively what he does. Moreover, that is exactly what he does in his computer code. His code does compute $1/2(ab+ba)$ by a computer simulation in which about half the time the answer is ab , and half the time it is ba . The average is therefore about $1/2(ab+ba)$. In fact the real part of this quaternion is exactly $1/2(ab+ba)$, the purely quaternionic part is small and is just statistical noise due to a finite sample.

RDG: I have added some remarks in the appendix about the reviewer's ideas. If the only thing wrong with Christian's work is poor notation or not quite correctly used terminology, then anyone should be able to write and publish their own exposition. I would happily collaborate with the reviewer on this enterprise, or give him any other support I can.

Appendix D

Report on Comment on “Quantum correlations are weaved by the spinors of the Euclidean primitives” by R.D. Gill, revision 1.

I thank the author for considering my comments carefully and overall I am happy with his modifications in revision 1. I therefore advise the accept the paper, provided the following minor language (and name) issues are addressed:

1. Page 1, line 21: George → **Anthony** (Lasenby). Same on page 4, line 31.
2. P3, 117: ... but they **are** also ...
3. P3, 131: ... which LR says **are** impossible ... (remove but and replace is → are)
4. P5, 136: ... **depends** only on the ...
5. P6, 110: ... at each trial **were** performed ...
6. P6, 123: ... that this **is** a fertile ...
7. P6, 156 ... ideas **needed** to ... (This is one way to fix this sentence, but the author may have other meanings in mind, so may fix it differently.)

Appendix E

Dear editors of RSOS

Thank you very much for review and the green light to proceed.

I believe I have taken account of all the comments of all reviewers and editors.

Below follows my response to all the points raised.

1. Please can the usage of 'fatal' in the abstract be removed and instead replaced with 'fundamental' or 'serious' (or similar)? This will help make the Comment more dispassionate (see bullet 3 below).

I have done this, not only in the abstract, but throughout the paper.

2. In the second paragraph of the Comment's introduction, please could you include additional references to link to earlier works that have critiqued Joy Christian's work; at present the author says he is one of many with concerns, but it is not immediately which other works have provided this criticism, and it would be helpful to readers to include additional references to other works to provide additional context.

I have added one explicit reference but have not listed the others. I have explained in the text that till recent years Christian's work did not get published in regular journals at all. Critics of his posted arXiv preprints, but since Christian's preprints did not achieve journal publication, there was, I can imagine, no point in submitting the criticisms. If the editors or referees are interested I could cite arXiv preprints by Moldoveanu, Grangier, and others, either in correspondence to you or in the paper itself.

3. In correspondence with the editorial office, the Editors have asked us to encourage the author to moderate the tone of the manuscript, please. They expressed a concern that the manuscript as written remains more emotive than would ordinarily be expected from a Comment assessing the scientific and mathematical merits of the original paper. A suggestion made by the Editors that may help with this is reducing usage of the original author's name and instead referring to them as 'the author'

I have taken up this suggestion.

4. A number of references are made to 'George Lasenby' in the text, when it appears that the author intends to refer to 'Anthony Lasenby' instead - we encourage you to doublecheck this.

Oops! I have double checked.

Reviewer: 5

Comments to the Author(s)

It should be Sabine, not Sabina (a few times in the appendix).

Corrected

Reviewer: 3

Comments to the Author(s)

Thank you for reviewing your manuscript in view of my comments. As a valuable contribution to this subject, I recommend its publication.

Thank you!

Reviewer: 2

Comments to the Author(s)

The present manuscript is a lucid and insightful piece of work. It clearly explains the mathematical core of Bell's inequality in its precise form. I concur with the author about Christian's errors in incorporating new definitions of locality and realism into a correct mathematical framework. The author has pointed out the algebra used in Christian's work is associative, but he should elucidate more on this point. Regarding the errors of the use of Geometric Algebra in the Christian's work, I concur with Lasenby, and the author has provided sufficient literature for it. I agree with the author about the inappropriateness of Christian's claim.

Thank you! I have slightly expanded my text at the place where the associativity is mentioned. Actually, it seemed to me that the difficulty here is that I was using both the words "multiplication" and "product". There is a multiplication operation. It takes two arguments, the result is called their product. An algebra is just a vector space with a compatible multiplication. I explained the meaning of the word compatible, and also wrote out the definition of associativity.

Reviewer: 4

Comments to the Author(s)

In the first round of review, I recommended the manuscript arguing that Bell and his followers as well as Bell-deniers are quite right in their beliefs as the so-called "Bell's theorem" in physics is, indeed, an undecidable hypothesis in mathematics. In my review I showed that the Copenhagen interpretation and its Born's rule leads to incompatibility and compatibility between quantum probability and classical probability at same time.

In this second round, the author included in his revision an Appendix with new arguments. I consider that the general aim of the debate about so-called "Bell's theorem" is a very valuable goal. I think, however, that the text needs to be improved before it is eventually published.

Thank you. You argued that Bell's theorem is an undecidable hypothesis in mathematics. I do not agree with your arguments, and I am extremely confident in this. I politely mentioned that I am not a specialist in the foundations of mathematics, but in this letter to you I will tell you that I have studied it intensively, starting from undergraduate courses given by J.H. Conway and frequent exposure over 45 years experience as a research mathematician with broad interests in all of mathematics.

I have done my best again to improve the text and to balance the wishes of some reviewers for more, and others for less material, of various kinds.

Reviewer: 4 (continued)

That said, let me examine some of the arguments included in Appendix:

- The sentence "A referee [Referee 1] suggests that the violation of Bell's theorem ..." in Appendix [line 40] needs to be changed for "A referee [Referee 4] suggests that the violation of Bell's theorem ..."

Thank you. I have avoided this problem by dropping the labels "Referee 1" and "Referee 4".

Reviewer: 4 (continued)

- In Appendix, the author wrote "I am not an expert but I admit that it is possible that the ZFC axioms are inconsistent, though in my opinion not likely. All this is however in my opinion irrelevant." The way he wrote suggests to us very strongly that his argument is passionate rather than scientific. These sentences are confusing, and it can harm the reading. How can the author say that "All this is however in my opinion irrelevant" if the author himself said that he is not an expert in the topic?

So, I suggest that the sentence "All this is however in my opinion irrelevant" must be suppressed to avoid cross purposes.

I have sharpened my claims. Though not a *specialist* in the foundations of mathematics, I do believe that I have enough expertise to make useful judgements here. Moreover, I have expanded my own arguments.

Reviewer: 4 (continued)

- The author also writes "I believe that this "foundations of mathematics" approaches is fruitless. It has been taken up a few times but gained no notable adherents."

These sentences must be rephrased because the so-called "Bell's theorem" has its substrate in Bell's inequalities, which are basic results from the Kolmogorov probability theory, which in turn is a result of set theory. As a consequence, Bell's theorem is directly linked to the foundations of mathematics. And, it is also important to note that an approach for QM based on the foundations of mathematics was proposed by Albert Einstein in his last work, then, the author's belief that such an approach gained no notable adherents is quite controversial.

Everything in mathematics is connected to the foundations of mathematics. Einstein was concerned about the application of mathematics to physics. Subtly different issues then arise. I have expanded my discussion on these points in the appendix.

Reviewer: 4 (continued)

The author mentions that "There are much more interesting off-beat solutions, such as Tim Palmer's and Sabina Hossenfelder's and Gerard't Hooft's ideas based on superdeterminism."

It is worth highlighting here that the idea of superdeterminism is grounded on a possible loophole in Bell's theorem, i.e., the possibility of there being a local probabilistic model on the Boolean domain (CHSH at most 2, or not greater than 2) compatible with a nonlocal probabilistic model on the continuum (CHSH greater than 2). But it is interesting that whenever a local probabilistic model compatible with quantum probabilities is constructed, a probabilistic model incompatible with quantum probabilities is constructed at the same time. This is superdeterminism!

I absolutely disagree with the reviewer's mathematical claims here (though without references, I may be misunderstanding him). Superdeterminism is a metaphysical loophole in Bell's theorem, not a mathematical loophole.

Reviewer: 4 (continued)

I recommend the manuscript for publication in RSOS, provided that the author considers the points listed above.

Thank you!

Reviewer: 6

Comments to the Author(s)

The paper doesn't contribute to knowledge. The author just criticizes and even in an offensive manner the Joy Christian work. It doesn't deserve to be published.

I point out errors in Joy Christian's work. I have done my best not to be offensive in any way.

Reviewer: 1

I thank the author for considering my comments carefully and overall I am happy with his modifications in revision 1. I therefore advise the accept the paper, provided the following minor language (and name) issues are addressed:

Page 1, line 21: George→Anthony (Lasenby). Same on page 4, line 31.

P3, l17: ... but they are also ...

P3, l31: ... which LR says are impossible ... (remove but and replace is → are)

P5, l36: ... depends only on the ...

P6, l10: ... at each trial were performed ...

P6, l23: ... that this is a fertile ...

P6, l56 ... ideas needed to ... (This is one way to fix this sentence, but the author may have other meanings in mind, so may fix it differently.)

Thank you very much! I have taken care of all these issues.

Appendix F**ROYAL SOCIETY
OPEN SCIENCE****Comment on "Quantum correlations are weaved by the
spinors of the Euclidean primitives"**

Journal:	Royal Society Open Science
Manuscript ID	RSOS-201909.R2
Article Type:	Comment
Date Submitted by the Author:	23-Oct-2021
Complete List of Authors:	Gill, Richard; Leiden University, Mathematics
Subject:	Quantum physics < PHYSICS, Applied mathematics < MATHEMATICS
Keywords:	quantum correlations, local causality, simulation, quaternions, octonions, geometric algebra
Subject Category:	Mathematics

ROYAL SOCIETY OPEN SCIENCE

rsos.royalsocietypublishing.org

Research

Article submitted to journal

Subject Areas:

quantum engineering/quantum physics

Keywords:

quantum correlations, local causality, Bell's theorem, spinors, quaternions, octonions

Author for correspondence:

Richard D. Gill

e-mail: gill@math.leidenuniv.nl

Comment on “Quantum correlations are weaved by the spinors of the Euclidean primitives”

R. D. Gill¹

¹Mathematical Institute, Leiden University, Netherlands

I point out fundamental mathematical errors in the recent paper published in this journal “Quantum correlations are weaved by the spinors of the Euclidean primitives” by Joy Christian.

1. Introduction

Christian (2018) [3], published in this journal *RSOS*, is perhaps the most ambitious of many papers by Joy Christian (in the sequel: “the author”) published between 2007 and 2021, all making the same assertion: quantum entanglement is not mysterious or weird or spooky. According to him, the correlations derived from it have a purely classical physical and “locally realistic” explanation. In each paper, he proposes a *local hidden variables model* (but not always the same one) which according to him explains those correlations and reproduces them. According to the celebrated result known as Bell's theorem [2], this is impossible. Christian argues that Bell's proof of his theorem is mathematically wrong. In the paper discussed here, [3], he claims that the three-dimensional space around us globally has the geometry of the 3-sphere S^3 : the surface of the unit ball in \mathbb{R}^4 . It is only locally flat. He connects this to special relativity, specifically to the solution of Einstein's field equations known as Friedmann-Robertson-Walker spacetime with a constant spatial curvature. He furthermore connects it to the 7-sphere S^7 , thought of as a quaternionic 3-sphere rather than a real 3-sphere.

© 2014 The Authors. Published by the Royal Society under the terms of the Creative Commons Attribution License <http://creativecommons.org/licenses/by/4.0/>, which permits unrestricted use, provided the original author and source are credited.

THE ROYAL SOCIETY
PUBLISHING

1
2
3
4
5
6
7
8
9
10
11
12
13
14
15
16
17
18
19
20
21
22
23
24
25
26
27
28
29
30
31
32
33
34
35
36
37
38
39
40
41
42
43
44
45
46
47
48
49
50
51
52
53
54
55
56
57
58
59
60

A constant theme in the author's papers is to exploit a very well known and very established part of mathematics, *Geometric Algebra* (GA). GA is based on the interplay between *Clifford Algebra* (CA), an important part of *abstract algebra*, and *Geometry*. It has been championed (by Anthony Lasenby, among others) as a universal language of physics. Important parts of quantum information theory have already been rewritten in the language of Geometric Algebra, though so far this did not have much impact on the field. My original interest in the author's work was precisely because, although the shortcomings (major conceptual and mathematical mistakes) were obvious, I wondered if his intuitions might have put him onto something.

In Gill (2020) [10] I published a critique of his series of papers as it stood then. Actually, my paper was by no means the first paper refuting many of his claims, but the first to look at their development over the years, and almost the first one to appear in a peer-reviewed journal. The exception was Weatherall (2013) [17], published in *Foundations of Physics*. Weatherall, like myself, saw that some useful morals could be drawn from the author's works since they did illustrate very common misconceptions about Bell's theorem. The author had not published any of his papers in peer reviewed journals, so many of the early critiques were not even submitted to regular journals, but remained, like his, as preprints on the preprint server *arXiv.org*. The author did respond vigorously to all critiques in yet more preprints and has always maintained that all criticism levelled at it had been mistaken and has been adequately refuted by himself.

In my paper, I argued that the author's work was built on a combination of ambiguous notation and elementary errors in logic, algebra and calculus. Nothing I said had not been said before, whether in *arXiv.org* preprints or on science-oriented social media such as *PubPeer* and on various blogs such as that of Scott Aaronson, or on the internet forum of the organisation FQXi. My 2020 paper included a section on an earlier version of the paper which is the topic of discussion here. The author's paper had briefly appeared in another journal and then been retracted.

Time did not stand still, and after 2018, the author published a pair of companion papers Christian (2019, 2020) in the journal *IEEE Access* [4], [5]. At the invitation of the editors of that journal I published a "Comment" Gill (2021) [11] on the second of the pair [5], and the author's (2021) "Reply" [6] has also already appeared. Another invited "Comment" by me [12] to the first of that pair has been submitted to *IEEE Access* and accepted subject to a last revision. I am grateful for the invitation by *RSOS* to react to [3] but in view of my earlier papers, I will keep this reaction very brief. I will focus on the mathematics, not on the physics, and refer to my earlier papers for the mathematical details. By the way, the algebraic core of the author's *RSOS* [3] paper was also analysed by Lasenby (2020) [16], who independently identified exactly the same problems which I had found.

If physicists want to "rescue" the author's ideas because they see something in the underlying idea, it is up to them to do so. Physicists have again and again enriched mathematics by intuitively and with deep physical insight discovering patterns and abstract structures hitherto not known to mathematicians. This will surely happen again and again in the future. Debate on Bell's theorem will also continue for many years to come. In an appendix I will discuss proposals by two referees of this paper to salvage the author's programme. I argue that they cannot succeed. In fact, there are several other controversial and, in my opinion physically more interesting options for "Bell-deniers". There are plenty of fora where these controversial alternative quantum foundations are vigorously discussed, and a huge (peer-reviewed) literature. I am afraid that the author's work has not added directly to the ongoing debate, though it did stimulate several original contributions which have made Bell's own case even stronger than it already was, and made the task of opponents thereby harder still.

In [6], the author finally admitted

That is not to say that Bell's theorem does not have a sound mathematical core. When stated as a mathematical theorem in probability theory, there can be no doubt about its validity. My work on the subject does not challenge this mathematical core, if it is viewed as a piece of mathematics.

This is however exactly what the author's works, and in particular his RSOS paper [3], do *not* do. He *does* challenge the core mathematics of Bell's work, and moreover, in [3], he builds this challenge on a new result of his own in pure mathematics (abstract algebra) which however is plain wrong. He went on in [6] to assert that what his work actually does is

challenge the metaphysical conclusions regarding locality and realism derived from that mathematical core. My work thus draws a sharp distinction between the mathematical core of Bell's theorem and the metaphysical conclusions derived from it.

But drawing that "essential distinction" is exactly what he does not do. He adopts Bell's own mathematical framing of the concepts of locality and realism and argues that one of the basic mathematical steps in Bell's proof is wrong. But it is his own argument which is evidently wrong. He does claim that our conception of space needs revision, but in view of its defects, his own work does not provide much support for that claim.

2. The Hurwitz theorem (algebra), Bell's theorem as probability theory, and Bell's theorem as computer science

In this section I discuss two established mathematical results which contradict *mathematical* claims in Christian (2018) [3]. These are: (a), the *Hurwitz theorem* (so called because it was conjectured by Hurwitz; it was only proved decades later) stating that \mathbb{R} , \mathbb{C} , \mathbb{H} and \mathbb{O} are the only four normed division algebras, see Baez (2002) [1]; and (b) the mathematical core of *Bell's theorem*, Bell (1964) [2] on the incompatibility of quantum mechanics (QM) with local realism (LR).

I also discuss two no-go theorems saying that certain quantum correlations cannot be simulated on a network of classical computers, where the allowed connections between computers mimic the spatial-temporal relations involved between physical subsystems of a typical Bell experiment. These can be thought of as theorems of computer science, but they are also merely repackagings of the mathematical core of Bell's theorem. I bring them up because the author illustrates his mathematical claims by Monte Carlo computer simulations, and I use them to provide more evidence for my own claims, that his theory is badly wrong.

These "theorems from computer science" (specialism: distributed computing) are (c) *Gull's theorem*, Gull (2016) [14], Bell's core result proved in a beautiful and original way using Fourier theory, and (d) a theorem of my own published in Gill (2003) [8], which is actually Bell's core result *enhanced* using martingale theory (probability) and randomisation (statistics).

Regarding Gull's theorem (c), astrophysicist Stephen Gull is another powerful promoter of Geometric Algebra. Readers familiar with Fourier series and time-series analysis might find Gull's proof of Bell's theorem much easier than Bell's. The fact that the result can be proven using a myriad different mathematical approaches is further evidence that it stands like a rock.

Regarding my own contribution (d), recent "loophole-free experimental violations of Bell inequalities", [15] and others, have shown that it is possible to exhibit in the quantum optics lab phenomena which QM predicts **but which but LR says** is impossible; phenomena which Einstein and others interpreted as "spooky action at a distance". This new generation of experiments is the most stringent ever. Alternative explanations of the observed correlations due to experimental shortcomings such as imperfect photo-detectors are ruled out. The problem is that experimental testing of Bell inequalities result in a finite amount of experimental data. Mathematical theorems about theoretical correlations in an ideal setting are not enough. My probabilistic result (d), in the form of an elegant refinement due to Hensen et al. (2015) [15], was used in the statistical analysis of the four celebrated loophole-free Bell experiments in 2015 and 2016 in Delft, Munich, Vienna and at NIST (Boulder, Colorado). Problems due to possible time variation, time trends, time dependence, and finite sample size have been ruled out.

Computer simulations of Bell-type experiments have become very popular, and here too, the same statistical issues arise. They are popular both among opponents and among supporters of

Bell. In the context of the author's work, some readers might imagine that thanks to mathematical subtleties which they cannot hope to appreciate, the author's work is still largely correct. An appeal to the Hurwitz theorem or to a mathematical version of Bell's theorem is an argument *ad verecundam*; by authority. Perhaps the author's work is merely blemished by details of notation or terminology. Fortunately for us, the author also presents computer simulations of his model. If that code faithfully represented his intended interpretation of his formulas, then by looking at the code one could deduce what he was trying to express in his published formulas. Moreover, one could check by inspection whether or not his code faithfully respects locality and realism. The simulations do seem to generate the results predicted by quantum mechanics. This means, in view of results (c) and/or (d), that it is not possible that the code satisfies the specifications agreed by Bell and the author.

It is indeed easy to check that the computer code effectively just draws the cosine curve built into the computer algebra package used by the author; it does not respect the constraints of local realism. It therefore weakens, rather than supports, his claims.

Were the author's claims correct, mathematics and science would be shaken to their very foundations. If his computer simulations were correct, everybody in the world with access to internet and a decent PC could have independently verified that he was right – quantum correlations do have a completely classical, local, explanation. All quantum computing hype would be destroyed. All physics textbooks would have to be rewritten. No establishment conspiracy could stop the news from getting out. But this has not happened. His claims are unfounded, and the supporting evidence which he offers is actually further evidence that he has not succeeded in converting possible physical insight into hard science.

(a) The Hurwitz theorem

The best reference for this theorem is John Baez' wonderful paper [1] on the octonions, which starts with excellent expository material. According to Baez's Theorem 1, the real numbers, the complex numbers, the quaternions, and the octonions are the only normed division algebras (up to isomorphism, of course). Baez' definition of a normed division algebra is a real vector space endowed with a compatible (i.e., bilinear, i.e., satisfying distributivity axioms) multiplication operation which we call a product, and a norm making it a normed vector space, such that the norm is moreover multiplicative: the norm of a product is the product of the norms. The multiplication operation itself need not be commutative or even associative. (Commutativity is the requirement $ab = ba$, associativity is $a(bc) = (ab)c$). The useful and important thing to know here is that a division algebra has no zero divisors. There do not exist elements A and B , neither equal to zero, such that $AB = 0$. However, the author's algebra has an element called the "pseudoscalar", I will denote it by M , such that $M^2 = 1$. It follows that $0 = M^2 - 1 = (M - 1)(M + 1)$. Taking norms, $0 = \|M - 1\| \cdot \|M + 1\|$. Hence $\|M - 1\| = 0$ or $\|M + 1\| = 0$. Therefore $M - 1 = 0$ or $M + 1 = 0$, which implies that $M = 1$ or $M = -1$. That is a contradiction. The author's algebra is associative, the octonions are not. It is known as $Cl_{(0,3)}(\mathbb{R})$, the well studied even sub-algebra of $Cl_{(4,0)}(\mathbb{R})$.

A clue that the author is not proficient in algebra is in plain view. He defines two algebras, built from two 8-dimensional real vector spaces \mathcal{K}^+ and \mathcal{K}^- by specifying a vector space basis for each algebra and multiplication tables for the 8 basis elements of each algebra. But they are the same algebra. The linear spans of those two bases are trivially the same. The multiplication operation is the same.

The author's claims regarding abstract algebra have naturally attracted the interest of algebraists. Anthony Lasenby, one of the founders of Geometric Algebra has published a paper Lasenby (2020) [16] detailing the errors.

The author seems to have been unaware of the Hurwitz theorem. He insists that his algebraic result is true and denies that it contradicts Hurwitz's theorem.

(b) Bell's theorem: the mathematical core

Bell (1964) [2], as rapidly improved by Clauser, Horne, Shimony and Holt (1969) [7], essentially proves the following theorem. Suppose that X_a and Y_b are a family of random variables on a single probability space, taking values in the set $\{-1, +1\}$, and where a and b denote directions in ordinary 3D Euclidean space, represented by unit vectors a, b . Then it is not possible that $\mathbb{E}(X_a Y_b) = -a \cdot b$ for all a and b .

Bell's proof works by focussing on two choices for a and two for b , delivering us four combinations of possible values of the pair (a, b) . Since four binary random variables are supported by a discrete probability space with just 16 elementary outcomes, a proof (using for instance the so-called CHSH inequality) can be framed in absolutely elementary terms [9]. No calculus is needed. No summation of infinite series. No knowledge of physics and in particular, no knowledge of quantum physics. (On a technical note: Bell's (1964) three correlation inequality is a corollary of the later four correlation inequality known as the CHSH-Bell inequality [7], which was later fully espoused by Bell himself.)

Bell used his mathematical result to argue that quantum mechanics violated the meta-physical principles of locality and realism. One can escape from this mathematical obstruction only by redefining the concepts of locality and realism. The author, however, does not take that well trodden route. He claims that the purely mathematical result is wrong. His alleged proof thereof is his explicit construction of a counterexample. I have elsewhere shown that his construction depends on simple errors in elementary algebra and calculus.

He also argues that Bell's proof contains a fundamental error in reasoning: the Bell-CHSH inequality involves correlations obtained from different sub-experiments involving measurements of non-commuting observables, and (he says) therefore cannot be combined. However, in quantum mechanics, even if two observables do not commute, a real linear combination of those observables is another observable. By the linearity encapsulated in the basic rules of quantum mechanics, expectation values of linear combinations of non-commuting observables are the same linear combination of the expectation values of each observable separately. If a local hidden variables model reproduces the statistical predictions of quantum mechanics, then it must reproduce this linearity.

(c) Gull's theorem

Consider a computer simulation of a Bell-CHSH type experiment. Initially, one could imagine three computers, a source computer sending information to two measurement locations, where two computers each simulate an apparatus which receives "stuff" from a source, and a "setting" (a measurement direction) supplied by an experimenter. The "measurement station computers" each output an "outcome" ± 1 . After a large number of trials, one collects the inputs (settings) and outputs ("outcomes") together and computes the correlation (meaning in this context, just the mean value of the product) between the outcomes, for each possible pair of input settings.

Now, if that could be done, one could also create a copy of the source measurement station, including all the data which is stored on it at the beginning of the computer simulation, and then merge each of the copies of the source with the two measurement stations, giving us together two *completely separated* classical computers, which perform the following task.

The two computers are both loaded with data and a computer program, and then disconnected. After that, in N rounds, each computer is supplied an input, and each computer supplies an output. After the complete run of N "trials" is completed, one collects all the data together, and correlates the outputs, for each possible pair of inputs. Gull [14] used to pose the question, as part of the "Part II" (i.e., third year undergraduate) exam in Theoretical Physics at Cambridge University: is it possible to recover, in the limit, the correlations $-a \cdot b$? He outlines the proof that this is impossible, using a really nice argument from Fourier analysis. Unfortunately he never bothered to publish a formal proof, we must make do with his overhead transparencies from a conference. Inspection of Gull's outline proof shows that he is pretty explicitly making a

particular “no use of memory” assumption. Each of those computer’s outputs, in trial n , depends only on the initial data stored in the computers and their programs, and on the new setting a or b , and on the trial number n , but not on the previous $n - 1$ inputs. With a student Dilara Karakozak, I have written out the mathematical details in Gill and Karakozak (2021) [13].

The author’s paper contains computer code: just one program which simulates many times two measurement settings and the value of some hidden variables which one can imagine created by nature in the source and transmitted to two measurement stations. Measurement outcomes are computed at each measurement station from the relevant setting and from the hidden variables. Next the program goes on to compute the correlation between measurement outcomes for any pair of measurement settings. This is where things go wrong: notice the code line `if (lambda==1) q=(NA NB) else q=(NB NA)`. One may check the algebra embodied in the code: the program simply computes $-a \cdot b$. It does not calculate the correlation between the actual ± 1 valued measurement outcomes! The program does not implement the mathematical model given in the paper.

(d) Gill’s theorem

My paper Gill (2003) [8] was my reaction to eminent scientists claiming that Bell’s theorem was false, and some of them claiming to even be able to prove this by simulated computer models, running on a distributed network of computers. Some moreover claimed that Bell had not taken account of “time” in his theory (not true, Bell discusses that explicitly in his works, though not perhaps in his famous first paper). In particular, I wanted to make a pretty secure bet with Luigi Accardi that this couldn’t be done. I was therefore concerned about inevitable statistical variation, and also by the possibility that the computer programs generating the n th pair of outcomes from the n th pair of settings might make use of information about the past $n - 1$ settings and outcomes. Using martingale theory I proved a probability bound showing that deviations from the Bell-CHSH inequality of any size had exponentially small probability, provided only that the binary setting choices at each trial were performed, outside of each measurement station, and again and again, by two fair coin tosses. The measurement station computers were even allowed to communicate with one another and with the source *between* each trial. This result was later refined and improved and used by all the experimenters in the four famous “loophole-free” Bell experiments of 2015, starting with the Delft experiment [15] and continuing with experiments in Vienna, at NIST in Boulder (Colorado), and in Munich. See [11] for further details.

3. Conclusion

The paper Christian (2018) [3] is irreparably flawed. The author wanted to provide theoretical underpinning to his physical intuition that quantum correlations are caused by the geometry of space, which, he suggests, is that of S^3 , not of \mathbb{R}^3 . He attempted to use Geometric Algebra for this task. I personally doubt that this is a fertile avenue for future research, but I am a mathematician, not a physicist. I would be delighted if anyone would prove me wrong. I do advise anyone interested in taking up the challenge to be well-informed as to the mathematical barriers embodied in the essential mathematical content at the core of Bell’s theorem.

Appendix

I here respond to comments of two referees.

A referee suggests that the violation of Bell’s theorem in experiment shows that mathematics is inconsistent. According to him, Bell’s theorem is both true and untrue. He points out problems with the ZFC axioms connected to infinite sets, and mentions that Einstein also intimated problems here for physics. Of course it is in principle possible that the ZFC axioms are inconsistent. Gödel proved that consistency can never be proved. All this is however in my opinion irrelevant. My paper [9] presents a “finitary” strengthening of the CHSH inequality – a

discrete probability, finite sample size version. As I mentioned in the paper, Bell-CHSH uses two settings on each side of the experiment. The hidden variables can be reduced to the four binary counterfactual measurement outcomes. We need a discrete probability space with just $2^4 = 16$ outcomes. There is no way that deep logical issues concerning the existence of real numbers would spoil the *mathematical* theorems I have discussed in this paper.

There are, in my opinion, more interesting off-beat solutions, such as Tim Palmer's and Sabine Hossenfelder's and Gerard 't Hooft's ideas based on superdeterminism. Palmer believes that fractal geometry and p-adic topology explain how nature can escape the constraints of Bell's theorem. Indeed, at the end of his life, Einstein speculatively raised metaphysical concerns about real numbers, and I see Palmer's approach as an exploration of those concerns. Explore different topologies, i.e., different conceptions of closeness. Sabine Hossenfelder argues that there is nothing necessarily "conspiratorial" in super-determinism; no need to delicately fine-tune parameters to make things come out exactly right. I think that she does not see the conspiratorial and unphysical ideas needed to imagine superdeterminism working its way all through a cascade of different kinds of random number generators used to generate measurement settings at two distant locations, in a subtle and perfect harmony with an underlying deterministic physics of lasers, photons and photo-detectors. I do not expect a satisfactory resolution of the problems facing those wanting to harmonise relativity and quantum theory and solve foundational issues in cosmology from this kind of approach, but it is good that these avenues are being explored.

Another referee suggests that the author's work is completely correct and that there are errors in my use of GA. He claims to have checked that the maths is correct using Matlab. Now it is true that many of the computations are locally correct. The referee says that the problem is that I do not acknowledge the author's new concepts of locality and realism and have not used his geometric setting. I disagree with this appraisal. The referee suggests that the only problem is that the author perhaps did not use entirely correct mathematical language. I suggest that the referee takes up the task of rewriting the author's model in unambiguous mathematical terms, and proceeds to implement the model faithfully in a Monte-Carlo computer simulation. He may also like to explain to the world what these new conceptions of locality and realism are in formal and unambiguous mathematical language. I would moreover be delighted if he would directly communicate with me so that we can try to thrash this out together.

Acknowledgements

Data Accessibility. This article has no additional data.

Authors' Contributions. This paper was written by myself alone.

Funding. RDG received no funding for his research.

Acknowledgements. RDG is grateful for the stimulating interactions with Dr Joy Christian over many years' lively debate.

References

1. Baez JC. 2002 The octonions. *emphBull. Amer. Math. Soc.* **39**, 145–205. <https://www.ams.org/journals/bull/2002-39-02/S0273-0979-01-00934-X>
2. Bell JS. (1964) On the Einstein Podolsky Rosen paradox. *Physics* **1**, 195–200. <https://journals.aps.org/ppf/abstract/10.1103/PhysicsPhysiqueFizika.1.195>
3. Christian J. 2018 Quantum correlations are weaved by the spinors of the Euclidean primitives. *R. Soc. Open Sci.* **5**, 180526 (40 pp.) <https://royalsocietypublishing.org/doi/full/10.1098/rsos.180526>
4. Christian J. 2019 Bell's theorem versus local realism in a quaternionic model of physical space. *IEEE Access* **7**, 133388–133409 <https://ieeexplore.ieee.org/document/8836453>

5. Christian J. 2020 Dr. Bertlmann's socks in a quaternionic world of ambidextral reality. *IEEE Access* **8**, 191028–191048. <https://ieeexplore.ieee.org/document/9226414>
6. Christian J. 2021 Reply to "Comment on 'Dr. Bertlmann's socks in a quaternionic world of ambidextral reality'". *IEEE Access* **9**, 72161 - 72171. <https://ieeexplore.ieee.org/document/9418997>
7. Clauser JF, Horne MA, Shimony A, Holt RA. 1969 Proposed experiment to test local hidden-variable theories. *Phys. Rev. Lett.* **23**, 880–884. <https://doi.org/10.1103/PhysRevLett.23.880>
8. Gill RD. 2003 Accardi contra Bell: the impossible coupling. pp. 133-154 in: Moore M, C. Leger C, Froda S. (eds.), *Mathematical Statistics and Applications: Festschrift for Constance van Eeden*, Lecture Notes–Monograph series, Institute of Mathematical Statistics, Hayward Ca. <https://doi.org/10.1214/lnms/1215091935>
9. Gill RD. 2014 Statistics, causality and Bell's theorem. *Statistical Science* **29** 512–528. <https://doi.org/10.1214/14-STS490>
10. Gill RD. 2020 Does Geometric Algebra provide a loophole to Bell's Theorem? *Entropy* **22**, 61 (21 pp.) <https://doi.org/10.3390/e22010061>
11. Gill RD. 2021 Comment on "Dr. Bertlmann's socks in a quaternionic world of ambidextral reality". *IEEE Access* **9**, 44592–44598. <https://ieeexplore.ieee.org/document/9380450>
12. Gill RD. 2021 Comment on "Bell's theorem versus local realism in a quaternionic model of physical space". Submitted to *IEEE Access*. <https://arxiv.org/abs/2103.00225v1>
13. Gill RD, Karakozak D. 2021 *Gull's theorem revisited*. Submitted. <https://arxiv.org/abs/2012.00719v5>
14. Gull S. 2016 *Quantum acausality and Bell's theorem*. Overhead transparencies, 4pp. <http://www.mrao.cam.ac.uk/~steve/maxent2009/images/bell.pdf>
15. Hensen B, Bernien H, Dréau A. et al. 2015 Loophole-free Bell inequality violation using electron spins separated by 1.3 kilometres. *Nature* **52**, 682–686. <https://doi.org/10.1038/nature15759>
16. Lasenby AN. 2020 A 1d Up approach to conformal geometric algebra: applications in line fitting and quantum mechanics. *Adv. Appl. Clifford Algebras* **30**, 16pp. <https://doi.org/10.1007/s00006-020-1046-0>
17. Weatherall JO. 2013 The Scope and Generality of Bell's Theorem. *Found. Phys* **43**, 1153–1169 (2013). <https://doi.org/10.1007/s10701-013-9737-1>

1
2
3
4
5
6
7
8
9
10
11
12
13
14
15
16
17
18
19
20
21
22
23
24
25
26
27
28
29
30
31
32
33
34
35
36
37
38
39
40
41
42
43
44
45
46
47
48
49
50
51
52
53
54
55
56
57
58
59
60

**ROYAL SOCIETY
OPEN SCIENCE**

rsos.royalsocietypublishing.org

Research

Article submitted to journal

Subject Areas:

quantum engineering/quantum physics

Keywords:

quantum correlations, local causality, Bell's theorem, spinors, quaternions, octonions

Author for correspondence:

Richard D. Gill

e-mail: gill@math.leidenuniv.nl

**THE ROYAL SOCIETY
PUBLISHING**

Comment on “Quantum correlations are weaved by the spinors of the Euclidean primitives”

R. D. Gill¹

¹Mathematical Institute, Leiden University, Netherlands

I point out fundamental mathematical errors in the recent paper published in this journal “Quantum correlations are weaved by the spinors of the Euclidean primitives” by Joy Christian.

1. Introduction

Christian (2018) [3], published in this journal *RSOS*, is perhaps the most ambitious of many papers by Joy Christian (in the sequel: “the author”) published between 2007 and 2021, all making the same assertion: quantum entanglement is not mysterious or weird or spooky. According to him, the correlations derived from it have a purely classical physical and “locally realistic” explanation. In each paper, he proposes a *local hidden variables model* (but not always the same one) which according to him explains those correlations and reproduces them. According to the celebrated result known as Bell's theorem [2], this is impossible. Christian argues that Bell's proof of his theorem is mathematically wrong. In the paper discussed here, [3], he claims that the three-dimensional space around us globally has the geometry of the 3-sphere S^3 : the surface of the unit ball in \mathbb{R}^4 . It is only locally flat. He connects this to special relativity, specifically to the solution of Einstein's field equations known as Friedmann-Robertson-Walker spacetime with a constant spatial curvature. He furthermore connects it to the 7-sphere S^7 , thought of as a quaternionic 3-sphere rather than a real 3-sphere.

© 2014 The Authors. Published by the Royal Society under the terms of the Creative Commons Attribution License <http://creativecommons.org/licenses/by/4.0/>, which permits unrestricted use, provided the original author and source are credited.

1
2
3
4
5
6
7
8
9
10
11
12
13
14
A constant theme in the author's papers is to exploit a very well known and very established part of mathematics, *Geometric Algebra* (GA). GA is based on the interplay between *Clifford Algebra* (CA), an important part of *abstract algebra*, and *Geometry*. It has been championed (by Anthony Lasenby, among others) as a universal language of physics. Important parts of quantum information theory have already been rewritten in the language of Geometric Algebra, though so far this did not have much impact on the field. My original interest in the author's work was precisely because, although the shortcomings (major conceptual and mathematical mistakes) were obvious, I wondered if his intuitions might have put him onto something.

15
16
17
18
19
20
21
22
23
24
In Gill (2020) [10] I published a critique of his series of papers as it stood then. **Actually, my paper was by no means the first paper refuting many of his claims, but the first to look at their development over the years, and almost the first one to appear in a peer-reviewed journal. The exception was Weatherall (2013) [17], published in *Foundations of Physics*. Weatherall, like myself, saw that some useful morals could be drawn from the author's works since they did illustrate very common misconceptions about Bell's theorem. The author had not published any of his papers in peer reviewed journals, so many of the early critiques were not even submitted to regular journals, but remained, like his, as preprints on the preprint server arXiv.org. The author did respond vigorously to all critiques in yet more preprints and has always maintained that all criticism levelled at it had been mistaken and has been adequately refuted by himself.**

25
26
27
28
29
30
In my paper, I argued that the author's work was built on a combination of ambiguous notation and elementary errors in logic, algebra and calculus. Nothing I said had not been said before, whether in arXiv.org preprints or on science-oriented social media such as PubPeer and on various blogs such as that of Scott Aaronson, or on the internet forum of the organisation FQXi. My 2020 paper included a section on an earlier version of the paper which is the topic of discussion here. The author's paper had briefly appeared in another journal and then been retracted.

31
32
33
34
35
36
37
38
39
40
41
Time did not stand still, and after 2018, the author published a pair of companion papers Christian (2019, 2020) in the journal *IEEE Access* [4], [5]. At the invitation of the editors of that journal I published a "Comment" Gill (2021) [11] on the second of the pair [5], and the author's (2021) "Reply" [6] has also already appeared. Another invited "Comment" by me [12] to the first of that pair **has been submitted to *IEEE Access* and accepted subject to a last revision.** I am grateful for the invitation by RSOS to react to [3] but in view of my earlier papers, I will keep this reaction very brief. I will focus on the mathematics, not on the physics, and refer to my earlier papers for the mathematical details. By the way, the algebraic core of the author's RSOS [3] paper was also analysed by Lasenby (2020) [16], who independently identified exactly the same problems which I had found.

42
43
44
45
46
47
48
49
50
51
52
53
If physicists want to "rescue" the author's ideas because they see something in the underlying idea, it is up to them to do so. Physicists have again and again enriched mathematics by intuitively and with deep physical insight discovering patterns and abstract structures hitherto not known to mathematicians. This will surely happen again and again in the future. Debate on Bell's theorem will also continue for many years to come. In an appendix I will discuss proposals by two referees of this paper to salvage the author's programme. I argue that they cannot succeed. In fact, there are several other controversial and, in my opinion physically more interesting options for "Bell-deniers". There are plenty of fora where these controversial alternative quantum foundations are vigorously discussed, and a huge (peer-reviewed) literature. I am afraid that the author's work has not added directly to the ongoing debate, though it did stimulate several original contributions which have made Bell's own case even stronger than it already was, and made the task of opponents thereby harder still.

54
55
In [6], the author finally admitted

56
57
58
59
60
That is not to say that Bell's theorem does not have a sound mathematical core. When stated as a mathematical theorem in probability theory, there can be no doubt about its validity. My work on the subject does not challenge this mathematical core, if it is viewed as a piece of mathematics.

This is however exactly what the author's works, and in particular his RSOS paper [3], do *not* do. He *does* challenge the core mathematics of Bell's work, and moreover, in [3], he builds this challenge on a new result of his own in pure mathematics (abstract algebra) which however is plain wrong. He went on in [6] to assert that what his work actually does is

challenge the metaphysical conclusions regarding locality and realism derived from that mathematical core. My work thus draws a sharp distinction between the mathematical core of Bell's theorem and the metaphysical conclusions derived from it.

But drawing that "essential distinction" is exactly what he does not do. He adopts Bell's own mathematical framing of the concepts of locality and realism and argues that one of the basic mathematical steps in Bell's proof is wrong. But it is his own argument which is evidently wrong. He does claim that our conception of space needs revision, but in view of its defects, his own work does not provide much support for that claim.

2. The Hurwitz theorem (algebra), Bell's theorem as probability theory, and Bell's theorem as computer science

In this section I discuss two established mathematical results which contradict *mathematical* claims in Christian (2018) [3]. These are: (a), the *Hurwitz theorem* (so called because it was conjectured by Hurwitz; it was only proved decades later) stating that \mathbb{R} , \mathbb{C} , \mathbb{H} and \mathbb{O} are the only four normed division algebras, see Baez (2002) [1]; and (b) the mathematical core of *Bell's theorem*, Bell (1964) [2] on the incompatibility of quantum mechanics (QM) with local realism (LR).

I also discuss two no-go theorems saying that certain quantum correlations cannot be simulated on a network of classical computers, where the allowed connections between computers mimic the spatial-temporal relations involved between physical subsystems of a typical Bell experiment. These can be thought of as theorems of computer science, but they are also merely repackagings of the mathematical core of Bell's theorem. I bring them up because the author illustrates his mathematical claims by Monte Carlo computer simulations, and I use them to provide more evidence for my own claims, that his theory is badly wrong.

These "theorems from computer science" (specialism: distributed computing) are (c) *Gull's theorem*, Gull (2016) [14], Bell's core result proved in a beautiful and original way using Fourier theory, and (d) a theorem of my own published in Gill (2003) [8], which is actually Bell's core result *enhanced* using martingale theory (probability) and randomisation (statistics).

Regarding Gull's theorem (c), astrophysicist Stephen Gull is another powerful promoter of Geometric Algebra. Readers familiar with Fourier series and time-series analysis might find Gull's proof of Bell's theorem much easier than Bell's. The fact that the result can be proven using a myriad different mathematical approaches is further evidence that it stands like a rock.

Regarding my own contribution (d), recent "loophole-free experimental violations of Bell inequalities", [15] and others, have shown that it is possible to exhibit in the quantum optics lab phenomena which QM predicts but which but LR says is impossible; phenomena which Einstein and others interpreted as "spooky action at a distance". This new generation of experiments is the most stringent ever. Alternative explanations of the observed correlations due to experimental shortcomings such as imperfect photo-detectors are ruled out. The problem is that experimental testing of Bell inequalities result in a finite amount of experimental data. Mathematical theorems about theoretical correlations in an ideal setting are not enough. My probabilistic result (d), in the form of an elegant refinement due to Hensen et al. (2015) [15], was used in the statistical analysis of the four celebrated loophole-free Bell experiments in 2015 and 2016 in Delft, Munich, Vienna and at NIST (Boulder, Colorado). Problems due to possible time variation, time trends, time dependence, and finite sample size have been ruled out.

Computer simulations of Bell-type experiments have become very popular, and here too, the same statistical issues arise. They are popular both among opponents and among supporters of

Bell. In the context of the author's work, some readers might imagine that thanks to mathematical subtleties which they cannot hope to appreciate, the author's work is still largely correct. An appeal to the Hurwitz theorem or to a mathematical version of Bell's theorem is an argument *ad verecundam*; by authority. Perhaps the author's work is merely blemished by details of notation or terminology. Fortunately for us, the author also presents computer simulations of his model. If that code faithfully represented his intended interpretation of his formulas, then by looking at the code one could deduce what he was trying to express in his published formulas. Moreover, one could check by inspection whether or not his code faithfully respects locality and realism. The simulations do seem to generate the results predicted by quantum mechanics. This means, in view of results (c) and/or (d), that it is not possible that the code satisfies the specifications agreed by Bell and the author.

It is indeed easy to check that the computer code effectively just draws the cosine curve built into the computer algebra package used by the author; it does not respect the constraints of local realism. It therefore weakens, rather than supports, his claims.

Were the author's claims correct, mathematics and science would be shaken to their very foundations. If his computer simulations were correct, everybody in the world with access to internet and a decent PC could have independently verified that he was right – quantum correlations do have a completely classical, local, explanation. All quantum computing hype would be destroyed. All physics textbooks would have to be rewritten. No establishment conspiracy could stop the news from getting out. But this has not happened. His claims are unfounded, and the supporting evidence which he offers is actually further evidence that he has not succeeded in converting possible physical insight into hard science.

(a) The Hurwitz theorem

The best reference for this theorem is John Baez' wonderful paper [1] on the octonions, which starts with excellent expository material. According to Baez's Theorem 1, the real numbers, the complex numbers, the quaternions, and the octonions are the only normed division algebras (up to isomorphism, of course). **Baez' definition of a normed division algebra is a real vector space endowed with a compatible (i.e., bilinear, i.e., satisfying distributivity axioms) multiplication operation which we call a product, and a norm making it a normed vector space, such that the norm is moreover multiplicative: the norm of a product is the product of the norms. The multiplication operation itself need not be commutative or even associative. (Commutativity is the requirement $ab = ba$, associativity is $a(bc) = (ab)c$). The useful and important thing to know here is that a division algebra has no zero divisors.** There do not exist elements A and B , neither equal to zero, such that $AB = 0$. However, the author's algebra has an element called the "pseudoscalar", I will denote it by M , such that $M^2 = 1$. It follows that $0 = M^2 - 1 = (M - 1)(M + 1)$. Taking norms, $0 = \|M - 1\| \cdot \|M + 1\|$. Hence $\|M - 1\| = 0$ or $\|M + 1\| = 0$. Therefore $M - 1 = 0$ or $M + 1 = 0$, which implies that $M = 1$ or $M = -1$. That is a contradiction. The author's algebra is associative, the octonions are not. It is known as $Cl_{(0,3)}(\mathbb{R})$, the well studied even sub-algebra of $Cl_{(4,0)}(\mathbb{R})$.

A clue that the author is not proficient in algebra is in plain view. He defines two algebras, built from two 8-dimensional real vector spaces \mathcal{K}^+ and \mathcal{K}^- by specifying a vector space basis for each algebra and multiplication tables for the 8 basis elements of each algebra. But they are the *same* algebra. The linear spans of those two bases are trivially the same. The multiplication operation is the same.

The author's claims regarding abstract algebra have naturally attracted the interest of algebraists. Anthony Lasenby, one of the founders of Geometric Algebra has published a paper Lasenby (2020) [16] detailing the errors.

The author seems to have been unaware of the Hurwitz theorem. He insists that his algebraic result is true and denies that it contradicts Hurwitz's theorem.

(b) Bell's theorem: the mathematical core

Bell (1964) [2], as rapidly improved by Clauser, Horne, Shimony and Holt (1969) [7], essentially proves the following theorem. Suppose that X_a and Y_b are a family of random variables on a single probability space, taking values in the set $\{-1, +1\}$, and where a and b denote directions in ordinary 3D Euclidean space, represented by unit vectors a, b . Then it is not possible that $\mathbb{E}(X_a Y_b) = -a \cdot b$ for all a and b .

Bell's proof works by focussing on two choices for a and two for b , delivering us four combinations of possible values of the pair (a, b) . Since four binary random variables are supported by a discrete probability space with just 16 elementary outcomes, a proof (using for instance the so-called CHSH inequality) can be framed in absolutely elementary terms [9]. No calculus is needed. No summation of infinite series. No knowledge of physics and in particular, no knowledge of quantum physics. (On a technical note: Bell's (1964) three correlation inequality is a corollary of the later four correlation inequality known as the CHSH-Bell inequality [7], which was later fully espoused by Bell himself.)

Bell used his mathematical result to argue that quantum mechanics violated the meta-physical principles of locality and realism. One can escape from this mathematical obstruction only by redefining the concepts of locality and realism. The author, however, does not take that well trodden route. He claims that the purely mathematical result is wrong. His alleged proof thereof is his explicit construction of a counterexample. I have elsewhere shown that his construction depends on simple errors in elementary algebra and calculus.

He also argues that Bell's proof contains a fundamental error in reasoning: the Bell-CHSH inequality involves correlations obtained from different sub-experiments involving measurements of non-commuting observables, and (he says) therefore cannot be combined. However, in quantum mechanics, even if two observables do not commute, a real linear combination of those observables is another observable. By the linearity encapsulated in the basic rules of quantum mechanics, expectation values of linear combinations of non-commuting observables are the same linear combination of the expectation values of each observable separately. If a local hidden variables model reproduces the statistical predictions of quantum mechanics, then it must reproduce this linearity.

(c) Gull's theorem

Consider a computer simulation of a Bell-CHSH type experiment. Initially, one could imagine three computers, a source computer sending information to two measurement locations, where two computers each simulate an apparatus which receives "stuff" from a source, and a "setting" (a measurement direction) supplied by an experimenter. The "measurement station computers" each output an "outcome" ± 1 . After a large number of trials, one collects the inputs (settings) and outputs ("outcomes") together and computes the correlation (meaning in this context, just the mean value of the product) between the outcomes, for each possible pair of input settings.

Now, if that could be done, one could also create a copy of the source measurement station, including all the data which is stored on it at the beginning of the computer simulation, and then merge each of the copies of the source with the two measurement stations, giving us together two *completely separated* classical computers, which perform the following task.

The two computers are both loaded with data and a computer program, and then disconnected. After that, in N rounds, each computer is supplied an input, and each computer supplies an output. After the complete run of N "trials" is completed, one collects all the data together, and correlates the outputs, for each possible pair of inputs. Gull [14] used to pose the question, as part of the "Part II" (i.e., third year undergraduate) exam in Theoretical Physics at Cambridge University: is it possible to recover, in the limit, the correlations $-a \cdot b$? He outlines the proof that this is impossible, using a really nice argument from Fourier analysis. Unfortunately he never bothered to publish a formal proof, we must make do with his overhead transparencies from a conference. Inspection of Gull's outline proof shows that he is pretty explicitly making a

particular “no use of memory” assumption. Each of those computer’s outputs, in trial n , depends only on the initial data stored in the computers and their programs, and on the new setting a or b , and on the trial number n , but not on the previous $n - 1$ inputs. With a student Dilara Karakozak, I have written out the mathematical details in Gill and Karakozak (2021) [13].

The author’s paper contains computer code: just one program which simulates many times two measurement settings and the value of some hidden variables which one can imagine created by nature in the source and transmitted to two measurement stations. Measurement outcomes are computed at each measurement station from the relevant setting and from the hidden variables. Next the program goes on to compute the correlation between measurement outcomes for any pair of measurement settings. This is where things go wrong: notice the code line `if (lambda==1) q=(NA NB) else q=(NB NA)`. One may check the algebra embodied in the code: the program simply computes $-a \cdot b$. It does not calculate the correlation between the actual ± 1 valued measurement outcomes! The program does not implement the mathematical model given in the paper.

(d) Gill’s theorem

My paper Gill (2003) [8] was my reaction to eminent scientists claiming that Bell’s theorem was false, and some of them claiming to even be able to prove this by simulated computer models, running on a distributed network of computers. Some moreover claimed that Bell had not taken account of “time” in his theory (not true, Bell discusses that explicitly in his works, though not perhaps in his famous first paper). In particular, I wanted to make a pretty secure bet with Luigi Accardi that this couldn’t be done. I was therefore concerned about inevitable statistical variation, and also by the possibility that the computer programs generating the n th pair of outcomes from the n th pair of settings might make use of information about the past $n - 1$ settings and outcomes. Using martingale theory I proved a probability bound showing that deviations from the Bell-CHSH inequality of any size had exponentially small probability, provided only that the binary setting choices at each trial were performed, outside of each measurement station, and again and again, by two fair coin tosses. The measurement station computers were even allowed to communicate with one another and with the source *between* each trial. This result was later refined and improved and used by all the experimenters in the four famous “loophole-free” Bell experiments of 2015, starting with the Delft experiment [15] and continuing with experiments in Vienna, at NIST in Boulder (Colorado), and in Munich. See [11] for further details.

3. Conclusion

The paper Christian (2018) [3] is irreparably flawed. The author wanted to provide theoretical underpinning to his physical intuition that quantum correlations are caused by the geometry of space, which, he suggests, is that of S^3 , not of \mathbb{R}^3 . He attempted to use Geometric Algebra for this task. I personally doubt that this is a fertile avenue for future research, but I am a mathematician, not a physicist. I would be delighted if anyone would prove me wrong. I do advise anyone interested in taking up the challenge to be well-informed as to the mathematical barriers embodied in the essential mathematical content at the core of Bell’s theorem.

Appendix

I here respond to comments of two referees.

A referee suggests that the violation of Bell’s theorem in experiment shows that mathematics is inconsistent. According to him, Bell’s theorem is both true and untrue. He points out problems with the ZFC axioms connected to infinite sets, and mentions that Einstein also intimated problems here for physics. **Of course it is in principle possible that the ZFC axioms are inconsistent. proved that consistency can never be proved. All this is however in my opinion irrelevant. My paper [9] presents a “finitary” strengthening of the CHSH inequality – a**

o

rsos.royalsocietypublishing.org R. Soc. open sci. 0000000

discrete probability, finite sample size version. As I mentioned in the paper, Bell-CHSH uses two settings on each side of the experiment. The hidden variables can be reduced to the four binary counterfactual measurement outcomes. We need a discrete probability space with just $2^4 = 16$ outcomes. There is no way that deep logical issues concerning the existence of real numbers would spoil the *mathematical* theorems I have discussed in this paper.

There are, in my opinion, more interesting off-beat solutions, such as Tim Palmer's and Sabine Hossenfelder's and Gerard 't Hooft's ideas based on superdeterminism. Palmer believes that fractal geometry and p-adic topology explain how nature can escape the constraints of Bell's theorem. Indeed, at the end of his life, Einstein speculatively raised metaphysical concerns about real numbers, and I see Palmer's approach as an exploration of those concerns. Explore different topologies, i.e., different conceptions of closeness. Sabine Hossenfelder argues that there is nothing necessarily "conspiratorial" in super-determinism; no need to delicately fine-tune parameters to make things come out exactly right. I think that she does not see the conspiratorial and unphysical ideas needed to imagine superdeterminism working its way all through a cascade of different kinds of random number generators used to generate measurement settings at two distant locations, in a subtle and perfect harmony with an underlying deterministic physics of lasers, photons and photo-detectors. I do not expect a satisfactory resolution of the problems facing those wanting to harmonise relativity and quantum theory and solve foundational issues in cosmology from this kind of approach, but it is good that these avenues are being explored.

Another referee suggests that the author's work is completely correct and that there are errors in my use of GA. He claims to have checked that the maths is correct using Matlab. Now it is true that many of the computations are locally correct. The referee says that the problem is that I do not acknowledge the author's new concepts of locality and realism and have not used his geometric setting. I disagree with this appraisal. The referee suggests that the only problem is that the author perhaps did not use entirely correct mathematical language. I suggest that the referee takes up the task of rewriting the author's model in unambiguous mathematical terms, and proceeds to implement the model faithfully in a Monte-Carlo computer simulation. He may also like to explain to the world what these new conceptions of locality and realism are in formal and unambiguous mathematical language. I would moreover be delighted if he would directly communicate with me so that we can try to thrash this out together.

Acknowledgements

Data Accessibility. This article has no additional data.

Authors' Contributions. This paper was written by myself alone.

Funding. RDG received no funding for his research.

Acknowledgements. RDG is grateful for the stimulating interactions with Dr Joy Christian over many years' lively debate.

References

1. Baez JC. 2002 The octonions. *emphBull. Amer. Math. Soc.* **39**, 145–205. <https://www.ams.org/journals/bull/2002-39-02/S0273-0979-01-00934-X>
2. Bell JS. (1964) On the Einstein Podolsky Rosen paradox. *Physics* **1**, 195–200. <https://journals.aps.org/ppf/abstract/10.1103/PhysicsPhysiqueFizika.1.195>
3. Christian J. 2018 Quantum correlations are weaved by the spinors of the Euclidean primitives. *R. Soc. Open Sci.* **5**, 180526 (40 pp.) <https://royalsocietypublishing.org/doi/full/10.1098/rsos.180526>
4. Christian J. 2019 Bell's theorem versus local realism in a quaternionic model of physical space. *IEEE Access* **7**, 133388–133409 <https://ieeexplore.ieee.org/document/8836453>

5. Christian J. 2020 Dr. Bertlmann's socks in a quaternionic world of ambidextral reality. *IEEE Access* **8**, 191028–191048. <https://ieeexplore.ieee.org/document/9226414>
6. Christian J. 2021 Reply to "Comment on 'Dr. Bertlmann's socks in a quaternionic world of ambidextral reality'". *IEEE Access* **9**, 72161 - 72171. <https://ieeexplore.ieee.org/document/9418997>
7. Clauser JF, Horne MA, Shimony A, Holt RA. 1969 Proposed experiment to test local hidden-variable theories. *Phys. Rev. Lett.* **23**, 880–884. <https://doi.org/10.1103/PhysRevLett.23.880>
8. Gill RD. 2003 Accardi contra Bell: the impossible coupling. pp. 133-154 in: Moore M, C. Leger C, Froda S. (eds.), *Mathematical Statistics and Applications: Festschrift for Constance van Eeden*, Lecture Notes–Monograph series, Institute of Mathematical Statistics, Hayward Ca. <https://doi.org/10.1214/lnms/1215091935>
9. Gill RD. 2014 Statistics, causality and Bell's theorem. *Statistical Science* **29** 512–528. <https://doi.org/10.1214/14-STS490>
10. Gill RD. 2020 Does Geometric Algebra provide a loophole to Bell's Theorem? *Entropy* **22**, 61 (21 pp.) <https://doi.org/10.3390/e22010061>
11. Gill RD. 2021 Comment on "Dr. Bertlmann's socks in a quaternionic world of ambidextral reality". *IEEE Access* **9**, 44592–44598. <https://ieeexplore.ieee.org/document/9380450>
12. Gill RD. 2021 Comment on "Bell's theorem versus local realism in a quaternionic model of physical space". Submitted to *IEEE Access*. <https://arxiv.org/abs/2103.00225v1>
13. Gill RD, Karakozak D. 2021 *Gull's theorem revisited*. Submitted. <https://arxiv.org/abs/2012.00719v5>
14. Gull S. 2016 *Quantum acausality and Bell's theorem*. Overhead transparencies, 4pp. <http://www.mrao.cam.ac.uk/~steve/maxent2009/images/bell.pdf>
15. Hensen B, Bernien H, Dréau A. *et al.* 2015 Loophole-free Bell inequality violation using electron spins separated by 1.3 kilometres. *Nature* **52**, 682–686. <https://doi.org/10.1038/nature15759>
16. Lasenby AN. 2020 A 1d Up approach to conformal geometric algebra: applications in line fitting and quantum mechanics. *Adv. Appl. Clifford Algebras* **30**, 16pp. <https://doi.org/10.1007/s00006-020-1046-0>
17. Weatherall JO. 2013 The Scope and Generality of Bell's Theorem. *Found. Phys* **43**, 1153–1169 (2013). <https://doi.org/10.1007/s10701-013-9737-1>

rsos.royalsocietypublishing.org

Research

Article submitted to journal

Subject Areas:

quantum engineering/quantum physics

Keywords:

quantum correlations, local causality, Bell's theorem, spinors, quaternions, octonions

Author for correspondence:

Richard D. Gill

e-mail: gill@math.leidenuniv.nl

Comment on “Quantum correlations are weaved by the spinors of the Euclidean primitives”

R. D. Gill¹

¹Mathematical Institute, Leiden University, Netherlands

I point out fundamental mathematical errors in the recent paper published in this journal “Quantum correlations are weaved by the spinors of the Euclidean primitives” by Joy Christian.

1. Introduction

Christian (2018) [3], published in this journal *RSOS*, is perhaps the most ambitious of many papers by Joy Christian (in the sequel: “the author”) published between 2007 and 2021, all making the same assertion: quantum entanglement is not mysterious or weird or spooky. According to him, the correlations derived from it have a purely classical physical and “locally realistic” explanation. In each paper, he proposes a *local hidden variables model* (but not always the same one) which according to him explains those correlations and reproduces them. According to the celebrated result known as Bell's theorem [2], this is impossible. Christian argues that Bell's proof of his theorem is mathematically wrong. In the paper discussed here, [3], he claims that the three-dimensional space around us globally has the geometry of the 3-sphere S^3 : the surface of the unit ball in \mathbb{R}^4 . It is only locally flat. He connects this to special relativity, specifically to the solution of Einstein's field equations known as Friedmann-Robertson-Walker spacetime with a constant spatial curvature. He furthermore connects it to the 7-sphere S^7 , thought of as a quaternionic 3-sphere rather than a real 3-sphere.

© 2014 The Authors. Published by the Royal Society under the terms of the Creative Commons Attribution License <http://creativecommons.org/licenses/by/4.0/>, which permits unrestricted use, provided the original author and source are credited.

A constant theme in the author's papers is to exploit a very well known and very established part of mathematics, *Geometric Algebra* (GA). GA is based on the interplay between *Clifford Algebra* (CA), an important part of *abstract algebra*, and *Geometry*. It has been championed (by Anthony Lasenby, among others) as a universal language of physics. Important parts of quantum information theory have already been rewritten in the language of Geometric Algebra, though so far this did not have much impact on the field. My original interest in the author's work was precisely because, although the shortcomings (major conceptual and mathematical mistakes) were obvious, I wondered if his intuitions might have put him onto something.

In Gill (2020) [10] I published a critique of his series of papers as it stood then. Actually, my paper was by no means the first paper refuting many of his claims, but the first to look at their development over the years, and almost the first one to appear in a peer-reviewed journal. The exception was Weatherall (2013) [17], published in *Foundations of Physics*. Weatherall, like myself, saw that some useful morals could be drawn from the author's works since they did illustrate very common misconceptions about Bell's theorem. The author had not published any of his papers in peer reviewed journals, so many of the early critiques were not even submitted to regular journals, but remained, like his, as preprints on the preprint server *arXiv.org*. The author did respond vigorously to all critiques in yet more preprints and has always maintained that all criticism levelled at it had been mistaken and has been adequately refuted by himself.

In my paper, I argued that the author's work was built on a combination of ambiguous notation and elementary errors in logic, algebra and calculus. Nothing I said had not been said before, whether in *arXiv.org* preprints or on science-oriented social media such as *PubPeer* and on various blogs such as that of Scott Aaronson, or on the internet forum of the organisation FQXi. My 2020 paper included a section on an earlier version of the paper which is the topic of discussion here. The author's paper had briefly appeared in another journal and then been retracted.

Time did not stand still, and after 2018, the author published a pair of companion papers Christian (2019, 2020) in the journal *IEEE Access* [4], [5]. At the invitation of the editors of that journal I published a "Comment" Gill (2021) [11] on the second of the pair [5], and the author's (2021) "Reply" [6] has also already appeared. Another invited "Comment" by me [12] to the first of that pair has been submitted to *IEEE Access* and accepted subject to a last revision. I am grateful for the invitation by *RSOS* to react to [3] but in view of my earlier papers, I will keep this reaction very brief. I will focus on the mathematics, not on the physics, and refer to my earlier papers for the mathematical details. By the way, the algebraic core of the author's *RSOS* [3] paper was also analysed by Lasenby (2020) [16], who independently identified exactly the same problems which I had found.

If physicists want to "rescue" the author's ideas because they see something in the underlying idea, it is up to them to do so. Physicists have again and again enriched mathematics by intuitively and with deep physical insight discovering patterns and abstract structures hitherto not known to mathematicians. This will surely happen again and again in the future. Debate on Bell's theorem will also continue for many years to come. In an appendix I will discuss proposals by two referees of this paper to salvage the author's programme. I argue that they cannot succeed. In fact, there are several other controversial and, in my opinion physically more interesting options for "Bell-deniers". There are plenty of fora where these controversial alternative quantum foundations are vigorously discussed, and a huge (peer-reviewed) literature. I am afraid that the author's work has not added directly to the ongoing debate, though it did stimulate several original contributions which have made Bell's own case even stronger than it already was, and made the task of opponents thereby harder still.

In [6], the author finally admitted

That is not to say that Bell's theorem does not have a sound mathematical core. When stated as a mathematical theorem in probability theory, there can be no doubt about its validity. My work on the subject does not challenge this mathematical core, if it is viewed as a piece of mathematics.

This is however exactly what the author's works, and in particular his RSOS paper [3], do *not* do. He *does* challenge the core mathematics of Bell's work, and moreover, in [3], he builds this challenge on a new result of his own in pure mathematics (abstract algebra) which however is plain wrong. He went on in [6] to assert that what his work actually does is

challenge the metaphysical conclusions regarding locality and realism derived from that mathematical core. My work thus draws a sharp distinction between the mathematical core of Bell's theorem and the metaphysical conclusions derived from it.

But drawing that "essential distinction" is exactly what he does not do. He adopts Bell's own mathematical framing of the concepts of locality and realism and argues that one of the basic mathematical steps in Bell's proof is wrong. But it is his own argument which is evidently wrong. He does claim that our conception of space needs revision, but in view of its defects, his own work does not provide much support for that claim.

2. The Hurwitz theorem (algebra), Bell's theorem as probability theory, and Bell's theorem as computer science

In this section I discuss two established mathematical results which contradict *mathematical* claims in Christian (2018) [3]. These are: (a), the *Hurwitz theorem* (so called because it was conjectured by Hurwitz; it was only proved decades later) stating that \mathbb{R} , \mathbb{C} , \mathbb{H} and \mathbb{O} are the only four normed division algebras, see Baez (2002) [1]; and (b) the mathematical core of *Bell's theorem*, Bell (1964) [2] on the incompatibility of quantum mechanics (QM) with local realism (LR).

I also discuss two no-go theorems saying that certain quantum correlations cannot be simulated on a network of classical computers, where the allowed connections between computers mimic the spatial-temporal relations involved between physical subsystems of a typical Bell experiment. These can be thought of as theorems of computer science, but they are also merely repackagings of the mathematical core of Bell's theorem. I bring them up because the author illustrates his mathematical claims by Monte Carlo computer simulations, and I use them to provide more evidence for my own claims, that his theory is badly wrong.

These "theorems from computer science" (specialism: distributed computing) are (c) *Gull's theorem*, Gull (2016) [14], Bell's core result proved in a beautiful and original way using Fourier theory, and (d) a theorem of my own published in Gill (2003) [8], which is actually Bell's core result *enhanced* using martingale theory (probability) and randomisation (statistics).

Regarding Gull's theorem (c), astrophysicist Stephen Gull is another powerful promoter of Geometric Algebra. Readers familiar with Fourier series and time-series analysis might find Gull's proof of Bell's theorem much easier than Bell's. The fact that the result can be proven using a myriad different mathematical approaches is further evidence that it stands like a rock.

Regarding my own contribution (d), recent "loophole-free experimental violations of Bell inequalities", [15] and others, have shown that it is possible to exhibit in the quantum optics lab phenomena which QM predicts but which but LR says is impossible; phenomena which Einstein and others interpreted as "spooky action at a distance". This new generation of experiments is the most stringent ever. Alternative explanations of the observed correlations due to experimental shortcomings such as imperfect photo-detectors are ruled out. The problem is that experimental testing of Bell inequalities result in a finite amount of experimental data. Mathematical theorems about theoretical correlations in an ideal setting are not enough. My probabilistic result (d), in the form of an elegant refinement due to Hensen et al. (2015) [15], was used in the statistical analysis of the four celebrated loophole-free Bell experiments in 2015 and 2016 in Delft, Munich, Vienna and at NIST (Boulder, Colorado). Problems due to possible time variation, time trends, time dependence, and finite sample size have been ruled out.

Computer simulations of Bell-type experiments have become very popular, and here too, the same statistical issues arise. They are popular both among opponents and among supporters of

Bell. In the context of the author's work, some readers might imagine that thanks to mathematical subtleties which they cannot hope to appreciate, the author's work is still largely correct. An appeal to the Hurwitz theorem or to a mathematical version of Bell's theorem is an argument *ad verecundam*; by authority. Perhaps the author's work is merely blemished by details of notation or terminology. Fortunately for us, the author also presents computer simulations of his model. If that code faithfully represented his intended interpretation of his formulas, then by looking at the code one could deduce what he was trying to express in his published formulas. Moreover, one could check by inspection whether or not his code faithfully respects locality and realism. The simulations do seem to generate the results predicted by quantum mechanics. This means, in view of results (c) and/or (d), that it is not possible that the code satisfies the specifications agreed by Bell *and* the author.

It is indeed easy to check that the computer code effectively just draws the cosine curve built into the computer algebra package used by the author; it does not respect the constraints of local realism. It therefore weakens, rather than supports, his claims.

Were the author's claims correct, mathematics and science would be shaken to their very foundations. If his computer simulations were correct, everybody in the world with access to internet and a decent PC could have independently verified that he was right – quantum correlations do have a completely classical, local, explanation. All quantum computing hype would be destroyed. All physics textbooks would have to be rewritten. No establishment conspiracy could stop the news from getting out. But this has not happened. His claims are unfounded, and the supporting evidence which he offers is actually further evidence that he has not succeeded in converting possible physical insight into hard science.

(a) The Hurwitz theorem

The best reference for this theorem is John Baez' wonderful paper [1] on the octonions, which starts with excellent expository material. According to Baez's Theorem 1, the real numbers, the complex numbers, the quaternions, and the octonions are the only normed division algebras (up to isomorphism, of course). Baez' definition of a normed division algebra is a real vector space endowed with a compatible (i.e., bilinear, i.e., satisfying distributivity axioms) multiplication operation which we call a product, and a norm making it a normed vector space, such that the norm is moreover multiplicative: the norm of a product is the product of the norms. The multiplication operation itself need not be commutative or even associative. (Commutativity is the requirement $ab = ba$, associativity is $a(bc) = (ab)c$). The useful and important thing to know here is that a division algebra has no zero divisors. There do not exist elements A and B , neither equal to zero, such that $AB = 0$. However, the author's algebra has an element called the "pseudoscalar", I will denote it by M , such that $M^2 = 1$. It follows that $0 = M^2 - 1 = (M - 1)(M + 1)$. Taking norms, $0 = \|M - 1\| \cdot \|M + 1\|$. Hence $\|M - 1\| = 0$ or $\|M + 1\| = 0$. Therefore $M - 1 = 0$ or $M + 1 = 0$, which implies that $M = 1$ or $M = -1$. That is a contradiction. The author's algebra is associative, the octonions are not. It is known as $Cl_{(0,3)}(\mathbb{R})$, the well studied even sub-algebra of $Cl_{(4,0)}(\mathbb{R})$.

A clue that the author is not proficient in algebra is in plain view. He defines two algebras, built from two 8-dimensional real vector spaces \mathcal{K}^+ and \mathcal{K}^- by specifying a vector space basis for each algebra and multiplication tables for the 8 basis elements of each algebra. But they are the *same* algebra. The linear spans of those two bases are trivially the same. The multiplication operation is the same.

The author's claims regarding abstract algebra have naturally attracted the interest of algebraists. Anthony Lasenby, one of the founders of Geometric Algebra has published a paper Lasenby (2020) [16] detailing the errors.

The author seems to have been unaware of the Hurwitz theorem. He insists that his algebraic result is true and denies that it contradicts Hurwitz's theorem.

(b) Bell's theorem: the mathematical core

Bell (1964) [2], as rapidly improved by Clauser, Horne, Shimony and Holt (1969) [7], essentially proves the following theorem. Suppose that X_a and Y_b are a family of random variables on a single probability space, taking values in the set $\{-1, +1\}$, and where a and b denote directions in ordinary 3D Euclidean space, represented by unit vectors a, b . Then it is not possible that $\mathbb{E}(X_a Y_b) = -a \cdot b$ for all a and b .

Bell's proof works by focussing on two choices for a and two for b , delivering us four combinations of possible values of the pair (a, b) . Since four binary random variables are supported by a discrete probability space with just 16 elementary outcomes, a proof (using for instance the so-called CHSH inequality) can be framed in absolutely elementary terms [9]. No calculus is needed. No summation of infinite series. No knowledge of physics and in particular, no knowledge of quantum physics. (On a technical note: Bell's (1964) three correlation inequality is a corollary of the later four correlation inequality known as the CHSH-Bell inequality [7], which was later fully espoused by Bell himself.)

Bell used his mathematical result to argue that quantum mechanics violated the meta-physical principles of locality and realism. One can escape from this mathematical obstruction only by redefining the concepts of locality and realism. The author, however, does not take that well trodden route. He claims that the purely mathematical result is wrong. His alleged proof thereof is his explicit construction of a counterexample. I have elsewhere shown that his construction depends on simple errors in elementary algebra and calculus.

He also argues that Bell's proof contains a fundamental error in reasoning: the Bell-CHSH inequality involves correlations obtained from different sub-experiments involving measurements of non-commuting observables, and (he says) therefore cannot be combined. However, in quantum mechanics, even if two observables do not commute, a real linear combination of those observables is another observable. By the linearity encapsulated in the basic rules of quantum mechanics, expectation values of linear combinations of non-commuting observables are the same linear combination of the expectation values of each observable separately. If a local hidden variables model reproduces the statistical predictions of quantum mechanics, then it must reproduce this linearity.

(c) Gull's theorem

Consider a computer simulation of a Bell-CHSH type experiment. Initially, one could imagine three computers, a source computer sending information to two measurement locations, where two computers each simulate an apparatus which receives "stuff" from a source, and a "setting" (a measurement direction) supplied by an experimenter. The "measurement station computers" each output an "outcome" ± 1 . After a large number of trials, one collects the inputs (settings) and outputs ("outcomes") together and computes the correlation (meaning in this context, just the mean value of the product) between the outcomes, for each possible pair of input settings.

Now, if that could be done, one could also create a copy of the source measurement station, including all the data which is stored on it at the beginning of the computer simulation, and then merge each of the copies of the source with the two measurement stations, giving us together two *completely separated* classical computers, which perform the following task.

The two computers are both loaded with data and a computer program, and then disconnected. After that, in N rounds, each computer is supplied an input, and each computer supplies an output. After the complete run of N "trials" is completed, one collects all the data together, and correlates the outputs, for each possible pair of inputs. Gull [14] used to pose the question, as part of the "Part II" (i.e., third year undergraduate) exam in Theoretical Physics at Cambridge University: is it possible to recover, in the limit, the correlations $-a \cdot b$? He outlines the proof that this is impossible, using a really nice argument from Fourier analysis. Unfortunately he never bothered to publish a formal proof, we must make do with his overhead transparencies from a conference. Inspection of Gull's outline proof shows that he is pretty explicitly making a

particular “no use of memory” assumption. Each of those computer’s outputs, in trial n , depends only on the initial data stored in the computers and their programs, and on the new setting a or b , and on the trial number n , but not on the previous $n - 1$ inputs. With a student Dilara Karakozak, I have written out the mathematical details in Gill and Karakozak (2021) [13].

The author’s paper contains computer code: just one program which simulates many times two measurement settings and the value of some hidden variables which one can imagine created by nature in the source and transmitted to two measurement stations. Measurement outcomes are computed at each measurement station from the relevant setting and from the hidden variables. Next the program goes on to compute the correlation between measurement outcomes for any pair of measurement settings. This is where things go wrong: notice the code line `if (lambda==1) q=(NA NB) else q=(NB NA)`. One may check the algebra embodied in the code: the program simply computes $-a \cdot b$. It does not calculate the correlation between the actual ± 1 valued measurement outcomes! The program does not implement the mathematical model given in the paper.

(d) Gill’s theorem

My paper Gill (2003) [8] was my reaction to eminent scientists claiming that Bell’s theorem was false, and some of them claiming to even be able to prove this by simulated computer models, running on a distributed network of computers. Some moreover claimed that Bell had not taken account of “time” in his theory (not true, Bell discusses that explicitly in his works, though not perhaps in his famous first paper). In particular, I wanted to make a pretty secure bet with Luigi Accardi that this couldn’t be done. I was therefore concerned about inevitable statistical variation, and also by the possibility that the computer programs generating the n th pair of outcomes from the n th pair of settings might make use of information about the past $n - 1$ settings and outcomes. Using martingale theory I proved a probability bound showing that deviations from the Bell-CHSH inequality of any size had exponentially small probability, provided only that the binary setting choices at each trial were performed, outside of each measurement station, and again and again, by two fair coin tosses. The measurement station computers were even allowed to communicate with one another and with the source *between* each trial. This result was later refined and improved and used by all the experimenters in the four famous “loophole-free” Bell experiments of 2015, starting with the Delft experiment [15] and continuing with experiments in Vienna, at NIST in Boulder (Colorado), and in Munich. See [11] for further details.

3. Conclusion

The paper Christian (2018) [3] is irreparably flawed. The author wanted to provide theoretical underpinning to his physical intuition that quantum correlations are caused by the geometry of space, which, he suggests, is that of S^3 , not of \mathbb{R}^3 . He attempted to use Geometric Algebra for this task. I personally doubt that this is a fertile avenue for future research, but I am a mathematician, not a physicist. I would be delighted if anyone would prove me wrong. I do advise anyone interested in taking up the challenge to be well-informed as to the mathematical barriers embodied in the essential mathematical content at the core of Bell’s theorem.

Appendix

I here respond to comments of two referees.

A referee suggests that the violation of Bell’s theorem in experiment shows that mathematics is inconsistent. According to him, Bell’s theorem is both true and untrue. He points out problems with the ZFC axioms connected to infinite sets, and mentions that Einstein also intimated problems here for physics. Of course it is in principle possible that the ZFC axioms are inconsistent. Gödel proved that consistency can never be proved. All this is however in my opinion irrelevant. My paper [9] presents a “finitary” strengthening of the CHSH inequality – a

discrete probability, finite sample size version. As I mentioned in the paper, Bell-CHSH uses two settings on each side of the experiment. The hidden variables can be reduced to the four binary counterfactual measurement outcomes. We need a discrete probability space with just $2^4 = 16$ outcomes. There is no way that deep logical issues concerning the existence of real numbers would spoil the *mathematical* theorems I have discussed in this paper.

There are, in my opinion, more interesting off-beat solutions, such as Tim Palmer's and Sabine Hossenfelder's and Gerard 't Hooft's ideas based on superdeterminism. Palmer believes that fractal geometry and p-adic topology explain how nature can escape the constraints of Bell's theorem. Indeed, at the end of his life, Einstein speculatively raised metaphysical concerns about real numbers, and I see Palmer's approach as an exploration of those concerns. Explore different topologies, i.e., different conceptions of closeness. Sabine Hossenfelder argues that there is nothing necessarily "conspiratorial" in super-determinism; no need to delicately fine-tune parameters to make things come out exactly right. I think that she does not see the conspiratorial and unphysical ideas needed to imagine superdeterminism working its way all through a cascade of different kinds of random number generators used to generate measurement settings at two distant locations, in a subtle and perfect harmony with an underlying deterministic physics of lasers, photons and photo-detectors. I do not expect a satisfactory resolution of the problems facing those wanting to harmonise relativity and quantum theory and solve foundational issues in cosmology from this kind of approach, but it is good that these avenues are being explored.

Another referee suggests that the author's work is completely correct and that there are errors in my use of GA. He claims to have checked that the maths is correct using Matlab. Now it is true that many of the computations are locally correct. The referee says that the problem is that I do not acknowledge the author's new concepts of locality and realism and have not used his geometric setting. I disagree with this appraisal. The referee suggests that the only problem is that the author perhaps did not use entirely correct mathematical language. I suggest that the referee takes up the task of rewriting the author's model in unambiguous mathematical terms, and proceeds to implement the model faithfully in a Monte-Carlo computer simulation. He may also like to explain to the world what these new conceptions of locality and realism are in formal and unambiguous mathematical language. I would moreover be delighted if he would directly communicate with me so that we can try to thrash this out together.

Acknowledgements

Data Accessibility. This article has no additional data.

Authors' Contributions. This paper was written by myself alone.

Funding. RDG received no funding for his research.

Acknowledgements. RDG is grateful for the stimulating interactions with Dr Joy Christian over many years' lively debate.

References

1. Baez JC. 2002 The octonions. *emphBull. Amer. Math. Soc.* **39**, 145–205.
<https://www.ams.org/journals/bull/2002-39-02/S0273-0979-01-00934-X>
2. Bell JS. (1964) On the Einstein Podolsky Rosen paradox. *Physics* **1**, 195–200. <https://journals.aps.org/ppf/abstract/10.1103/PhysicsPhysiqueFizika.1.195>
3. Christian J. 2018 Quantum correlations are weaved by the spinors of the Euclidean primitives. *R. Soc. Open Sci.* **5**, 180526 (40 pp.)
<https://royalsocietypublishing.org/doi/full/10.1098/rsos.180526>
4. Christian J. 2019 Bell's theorem versus local realism in a quaternionic model of physical space. *IEEE Access* **7**, 133388–133409
<https://ieeexplore.ieee.org/document/8836453>

5. Christian J. 2020 Dr. Bertlmann's socks in a quaternionic world of ambidextral reality. *IEEE Access* **8**, 191028–191048. <https://ieeexplore.ieee.org/document/9226414>
6. Christian J. 2021 Reply to "Comment on "Dr. Bertlmann's socks in a quaternionic world of ambidextral reality"". *IEEE Access* **9**, 72161 - 72171. <https://ieeexplore.ieee.org/document/9418997>
7. Clauser JF, Horne MA, Shimony A, Holt RA. 1969 Proposed experiment to test local hidden-variable theories. *Phys. Rev. Lett.* **23**, 880–884. <https://doi.org/10.1103/PhysRevLett.23.880>
8. Gill RD. 2003 Accardi contra Bell: the impossible coupling. pp. 133-154 in: Moore M, C. Leger C, Froda S. (eds.), *Mathematical Statistics and Applications: Festschrift for Constance van Eeden*, Lecture Notes–Monograph series, Institute of Mathematical Statistics, Hayward Ca. <https://doi.org/10.1214/lnms/1215091935>
9. Gill RD. 2014 Statistics, causality and Bell's theorem. *Statistical Science* **29** 512–528. <https://doi.org/10.1214/14-STS490>
10. Gill RD. 2020 Does Geometric Algebra provide a loophole to Bell's Theorem? *Entropy* **22**, 61 (21 pp.) <https://doi.org/10.3390/e22010061>
11. Gill RD. 2021 Comment on "Dr. Bertlmann's socks in a quaternionic world of ambidextral reality". *IEEE Access* **9**, 44592–44598. <https://ieeexplore.ieee.org/document/9380450>
12. Gill RD. 2021 Comment on "Bell's theorem versus local realism in a quaternionic model of physical space". Submitted to *IEEE Access*. <https://arxiv.org/abs/2103.00225v1>
13. Gill RD, Karakozak D. 2021 *Gull's theorem revisited*. Submitted. <https://arxiv.org/abs/2012.00719v5>
14. Gull S. 2016 *Quantum acausality and Bell's theorem*. Overhead transparencies, 4pp. <http://www.mrao.cam.ac.uk/~steve/maxent2009/images/bell.pdf>
15. Hensen B, Bernien H, Dréau A. *et al.* 2015 Loophole-free Bell inequality violation using electron spins separated by 1.3 kilometres. *Nature* **52**, 682–686. <https://doi.org/10.1038/nature15759>
16. Lasenby AN. 2020 A 1d Up approach to conformal geometric algebra: applications in line fitting and quantum mechanics. *Adv. Appl. Clifford Algebras* **30**, 16pp. <https://doi.org/10.1007/s00006-020-1046-0>
17. Weatherall JO. 2013 The Scope and Generality of Bell's Theorem. *Found. Phys* **43**, 1153–1169 (2013). <https://doi.org/10.1007/s10701-013-9737-1>

1
2 Dear editors of RSOS

3
4 Thank you very much for review and the green light to proceed.

5
6 I believe I have taken account of all the comments of all reviewers and editors.

7
8 Below follows my response to all the points raised.

9
10 **1. Please can the usage of 'fatal' in the abstract be removed and instead replaced with 'fundamental' or**
11 **'serious' (or similar)? This will help make the Comment more dispassionate (see bullet 3 below).**

12
13 I have done this, not only in the abstract, but throughout the paper.

14
15
16 **2. In the second paragraph of the Comment's introduction, please could you include additional**
17 **references to link to earlier works that have critiqued Joy Christian's work; at present the author says**
18 **he is one of many with concerns, but it is not immediately which other works have provided this**
19 **criticism, and it would be helpful to readers to include additional references to other works to provide**
20 **additional context.**

21
22
23 I have added one explicit reference but have not listed the others. I have explained in the text that till recent
24 years Christian's work did not get published in regular journals at all. Critics of his posted arXiv preprints,
25 but since Christian's preprints did not achieve journal publication, there was, I can imagine, no point in
26 submitting the criticisms. If the editors or referees are interested I could cite arXiv preprints by Moldoveanu,
27 Grangier, and others, either in correspondence to you or in the paper itself.

28
29
30 **3. In correspondence with the editorial office, the Editors have asked us to encourage the author to**
31 **moderate the tone of the manuscript, please. They expressed a concern that the manuscript as written**
32 **remains more emotive than would ordinarily be expected from a Comment assessing the scientific and**
33 **mathematical merits of the original paper. A suggestion made by the Editors that may help with this is**
34 **reducing usage of the original author's name and instead referring to them as 'the author'**

35
36 I have taken up this suggestion.

37
38
39 **4. A number of references are made to 'George Lasenby' in the text, when it appears that the author**
40 **intends to refer to 'Anthony Lasenby' instead - we encourage you to doublecheck this.**

41
42
43 Oops! I have double checked.

44
45
46 **Reviewer: 5**
47 **Comments to the Author(s)**
48 **It should be Sabine, not Sabina (a few times in the appendix).**

49
50 Corrected

51
52
53 **Reviewer: 3**
54 **Comments to the Author(s)**
55 **Thank you for reviewing your manuscript in view of my comments. As a valuable contribution to this**
56 **subject, I recommend its publication.**

57
58 Thank you!

59
60

1
2
3 **Reviewer: 2**

4 **Comments to the Author(s)**

5 **The present manuscript is a lucid and insightful piece of work. It clearly explains the mathematical**
6 **core of Bell's inequality in its precise form. I concur with the author about Christian's errors in**
7 **incorporating new definitions of locality and realism into a correct mathematical framework. The**
8 **author has pointed out the algebra used in Christian's work is associative, but he should elucidate**
9 **more on this point. Regarding the errors of the use of Geometric Algebra in the Christian's work, I**
10 **concur with Lasenby, and the author has provided sufficient literature for it. I agree with the author**
11 **about the inappropriateness of Christian's claim.**

12
13
14
15
16 Thank you! I have slightly expanded my text at the place where the associativity is mentioned. Actually, it
17 seemed to me that the difficulty here is that I was using both the words "multiplication" and "product". There
18 is a multiplication operation. It takes two arguments, the result is called their product. An algebra is just a
19 vector space with a compatible multiplication. I explained the meaning of the word compatible, and also
20 wrote out the definition of associativity.

21
22
23 **Reviewer: 4**

24 **Comments to the Author(s)**

25 **In the first round of review, I recommended the manuscript arguing that Bell and his followers as well**
26 **as Bell-deniers are quite right in their beliefs as the so-called "Bell's theorem" in physics is, indeed, an**
27 **undecidable hypothesis in mathematics. In my review I showed that the Copenhagen interpretation**
28 **and its Born's rule leads to incompatibility and compatibility between quantum probability and**
29 **classical probability at same time.**

30 **In this second round, the author included in his revision an Appendix with new arguments. I consider**
31 **that the general aim of the debate about so-called "Bell's theorem" is a very valuable goal. I think,**
32 **however, that the text needs to be improved before it is eventually published.**

33
34
35 Thank you. You argued that Bell's theorem is an undecidable hypothesis in mathematics. I do not agree with
36 your arguments, and I am extremely confident in this. I politely mentioned that I am not a specialist in the
37 foundations of mathematics, but in this letter to you I will tell you that I have studied it intensively, starting
38 from undergraduate courses given by J.H. Conway and frequent exposure over 45 years experience as a
39 research mathematician with broad interests in all of mathematics.

40
41 I have done my best again to improve the text and to balance the wishes of some reviewers for more, and
42 others for less material, of various kinds.

43
44
45
46 **Reviewer: 4 (continued)**

47 **That said, let me examine some of the arguments included in Appendix:**

48 **- The sentence "A referee [Referee 1] suggests that the violation of Bell's theorem ..." in Appendix [line**
49 **40] needs to be changed for "A referee [Referee 4] suggests that the violation of Bell's theorem ..."**

50
51
52
53 Thank you. I have avoided this problem by dropping the labels "Referee 1" and "Referee 4".

54
55
56
57 **Reviewer: 4 (continued)**

58 **- In Appendix, the author wrote "I am not an expert but I admit that it is possible that the ZFC**
59 **axioms are inconsistent, though in my opinion not likely. All this is however in my opinion irrelevant."**
60 **The way he wrote suggests to us very strongly that his argument is passionate rather than scientific.**

1
2 **These sentences are confusing, and it can harm the reading. How can the author say that "All this is**
3 **however in my opinion irrelevant" if the author himself said that he is not an expert in the topic?**
4 **So, I suggest that the sentence "All this is however in my opinion irrelevant" must be suppressed to**
5 **avoid cross purposes.**
6

7 I have sharpened my claims. Though not a *specialist* in the foundations of mathematics, I do believe that I
8 have enough expertise to make useful judgements here. Moreover, I have expanded my own arguments.
9

10
11
12 **Reviewer: 4 (continued)**

13 - **The author also writes "I believe that this "foundations of mathematics" approaches is fruitless. It**
14 **has been taken up a few times but gained no notable adherents."**

15 **These sentences must be rephrased because the so-called "Bell's theorem" has its substrate in Bell's**
16 **inequalities, which are basic results from the Kolmogorov probability theory, which in turn is a result**
17 **of set theory. As a consequence, Bell's theorem is directly linked to the foundations of mathematics.**
18 **And, it is also important to note that an approach for QM based on the foundations of mathematics**
19 **was proposed by Albert Einstein in his last work, then, the author's belief that such an approach**
20 **gained no notable adherents is quite controversial.**
21
22

23
24 Everything in mathematics is connected to the foundations of mathematics. Einstein was concerned about the
25 application of mathematics to physics. Subtly different issues then arise. I have expanded my discussion on
26 these points in the appendix.
27
28

29
30 **Reviewer: 4 (continued)**

31 **The author mentions that "There are much more interesting off-beat solutions, such as Tim Palmer's**
32 **and Sabina Hossenfelder's and Gerard't Hooft's ideas based on superdeterminism."**

33 **It is worth highlighting here that the idea of superdeterminism is grounded on a possible loophole in**
34 **Bell's theorem, i.e., the possibility of there being a local probabilistic model on the Boolean domain**
35 **(CHSH at most 2, or not greater than 2) compatible with a nonlocal probabilistic model on the**
36 **continuum (CHSH greater than 2). But it is interesting that whenever a local probabilistic model**
37 **compatible with quantum probabilities is constructed, a probabilistic model incompatible with**
38 **quantum probabilities is constructed at the same time. This is superdeterminism!**
39
40

41
42 I absolutely disagree with the reviewer's mathematical claims here (though without references, I may be
43 misunderstanding him). Superdeterminism is a metaphysical loophole in Bell's theorem, not a mathematical
44 loophole.
45
46

47
48 **Reviewer: 4 (continued)**

49 **I recommend the manuscript for publication in RSOS, provided that the author considers the points**
50 **listed above.**
51

52 Thank you!
53
54

55 **Reviewer: 6**

56 **Comments to the Author(s)**

57 **The paper doesn't contribute to knowledge. The author just criticizes and even in an offensive manner**
58 **the Joy Christian work. It doesn't deserve to be published.**
59
60

1
2 I point out errors in Joy Christian's work. I have done my best not to be offensive in any way.
3
4
5

6 **Reviewer: 1**

7 **I thank the author for considering my comments carefully and overall I am happy with his**
8 **modifications in revision 1. I therefore advise the accept the paper, provided the following minor**
9 **language (and name) issues are addressed:**

10 **Page 1, line 21: George→Anthony (Lasenby). Same on page 4, line 31.**

11 **P3, l17: ... but they are also ...**

12 **P3, l31: ... which LR says are impossible ... (remove but and replace is → are)**

13 **P5, l36: ... depends only on the ...**

14 **P6, l10: ... at each trial were performed ...**

15 **P6, l23: ... that this is a fertile ...**

16 **P6, l56 ... ideas needed to ... (This is one way to fix this sentence, but the author may have other**
17 **meanings in mind, so may fix it differently.)**

18
19
20
21 Thank you very much! I have taken care of all these issues.
22
23
24
25
26
27
28
29
30
31
32
33
34
35
36
37
38
39
40
41
42
43
44
45
46
47
48
49
50
51
52
53
54
55
56
57
58
59
60

Appendix G

Dear editors

Thank you very much for the positive review. I have carried out all the requests of the editors and the reviewers except for one referee, whose arguments I simply do not agree with. That is Reviewer 4, who has an utterly off-beat point of view, namely that Bell's theorem is both true and not true, and that mathematics as we know it is inconsistent.

Below are my detailed responses, and I have prepared both a clean pdf of the new revision as well as one with all changes highlighted.

Yours
Richard Gill

Associate Editor: 1

The paper contains a number of loose statements of the type that are verboten in a scholarly article. Please refer to the attachment called "RSOS-201909.R2_annotated.pdf". There you will find a pdf of your paper with detailed annotations for correction.

Author response

I have taken account of all your comments

Author action

All recommended corrections and deletions have been carried out

Reviewer: 1

Please see one phrase highlighted and commented in the manuscript. (See attached "RSOS-201909.R2_with one comment on page 4.pdf")

Author response

Sorry that I missed this correction the last time round

Author action

I have been rewritten the sentence

Reviewer: 5

The answers to the referees are lengthy and not so informative. I would suggest to shorten them or leave them away.

Author response

Other reviewers wanted to have them and it is editorial policy to include such a section. The section is now shorter still thanks to deletions of some superfluous remarks.

Author action

Some shortening has been done

Reviewer: 2

The author has incorporated my suggestions as well as the concerns of other reviewers in the revised version. I recommend publication.

Author response

Thank you!

Author action

No action

Reviewer: 4

It is important to make clear for the broader public that so-called "Bell's theorem" states that a nonlocal model of probability - such as that given by square of sine function (Born rule) - has nothing to do with a local model of probability distribution. I showed this in detail in my report above for a case of three coins tossed simultaneously. So, the so-called "Bell's theorem" merely states that continuous sample spaces are incompatible with discrete (Boolean) sample spaces. [Just one remark about Born rule: considering the Copenhagen interpretation, the predictions of quantum mechanics can be given by the Born rule].

However, I also showed in my report that the Born rule can be directly obtained from a local model of probability distribution. Hence, the continuous sample space are compatible with discrete sample spaces.

That is a paradox, and means that proving or disproving anything about Bell's statement within ZFC is impossible. The theory has a loophole (super-determinism), and as a result, the so-called "Bell's theorem" is, indeed, the very continuum hypothesis (an undecidable hypothesis), according to the references I mentioned in my report.

It's purely math!

As arguments and counterarguments regarding the so-called "Bell's theorem" are true, then I recommend the publication of the manuscript, once there cannot be an answer to the problem.

Author response

I'm afraid I totally disagree with you. I found your mathematical analysis unconvincing. I do not agree that there is some paradox here connected to inadequacy of ZFC. The paper already contained my argument for that, in the discussion section. You have not responded to my counterarguments, so there is nothing more I can say about this.

Author action

No action